# Contextual Similarity Distillation: Ensemble Uncertainties with a Single Model

**Moritz A. Zanger, Pascal R. Van der Vaart, Wendelin Böhmer & Matthijs T. J. Spaan**
Department of Intelligent Systems, Delft University of Technology
Delft, 2628 XE, The Netherlands
Correspondence to: `m.a.zanger@tudelft.nl`

## Abstract

Uncertainty quantification is a critical aspect of reinforcement learning and deep learning, with numerous applications ranging from efficient exploration and stable offline reinforcement learning to outlier detection in medical diagnostics. The scale of modern neural networks, however, complicates the use of many theoretically well-motivated approaches such as full Bayesian inference. Approximate methods like deep ensembles can provide reliable uncertainty estimates but still remain computationally expensive. In this work, we propose contextual similarity distillation, a novel approach that explicitly estimates the variance of an ensemble of deep neural networks with a single model, without ever learning or evaluating such an ensemble in the first place. Our method builds on the predictable learning dynamics of wide neural networks, governed by the neural tangent kernel, to derive an efficient approximation of the predictive variance of an infinite ensemble. Specifically, we reinterpret the computation of ensemble variance as a supervised regression problem with kernel similarities as regression targets. The resulting model can estimate predictive variance at inference time with a single forward pass, and can make use of unlabeled target-domain data or data augmentations to refine its uncertainty estimates. We empirically validate our method across a variety of out-of-distribution detection benchmarks and sparse-reward reinforcement learning environments. We find that our single-model method performs competitively and sometimes superior to ensemble-based baselines and serves as a reliable signal for efficient exploration. These results, we believe, position contextual similarity distillation as a principled and scalable alternative for uncertainty quantification in reinforcement learning and general deep learning.

## 1 Introduction

With the deployment of increasingly large deep learning systems to real-world applications, efficient uncertainty quantification has become an essential challenge of modern deep learning. Assessing the reliability in predictions is crucial in applications ranging from out-of-distribution (OOD) detection to deep reinforcement learning (RL), where uncertainty estimation is used to drive exploration, stabilize offline learning, increase data efficiency, or to design cautious, safety-aware agents. A necessary condition for designing and deploying such agents is their ability to quantify uncertainty reliably and efficiently.

Bayesian methods for deep neural networks address this challenge with a solid theoretical footing (Goan and Fookes, 2020; Pearce et al., 2020; Izmailov et al., 2021) but often require coarse approximations or costly sampling from a complex posterior. To this end, deep ensembles from random initializations Lakshminarayanan et al. (2017); Osband et al. (2016); Qin et al. (2022) have emerged as a simple but reliable method for estimating predictive uncertainty in neural networks. While usually more efficient than full Bayesian inference, the computational cost of training several models remains a burden, particularly with increasing parameter spaces.

In this paper, we introduce contextual similarity distillation (CSD), a novel single-model approach that directly estimates the variance of a random initialization ensemble of deep NNs without ever training or evaluating such an ensemble in the first place. The theoretical motivation for our approach

is derived from recent work characterizing the learning dynamics of wide neural networks through the Neural Tangent Kernel (NTK, Jacot et al., 2018; Lee et al., 2020). Under some conditions, this setting allows us to describe deep ensembles and in particular their predictive variance by the NTK Gaussian Process (NTK GP, He et al., 2020), providing an analytical expression for ensemble uncertainties. Although one can in principle solve these analytical expressions explicitly without requiring training of an ensemble of models, these computations quickly become infeasible when considering large models or datasets, as frequently encountered in the field of RL.

In contrast, we devise a novel method called contextual similarity distillation (CSD) that is amenable to regular training pipelines based on gradient descent and approximates predictive ensemble variance with a single forward pass. We derive our method from the insight that ensemble variance can be obtained as the result of a structured supervised regression problem, where labels correspond to kernel similarities between training points and a test point $x_t$. As a result, one can obtain the predictive variance of a deep ensemble for a known query point $x_t$ by training a single NN on a regression task using gradient descent and a carefully designed label function dependent on $x_t$. We then extend this "single-query" approach to work efficiently for arbitrary queries $x_t$ by formulating a contextualized regression model that involves regression tasks with a family of context-dependent label functions. This formulation moreover enables CSD to refine its uncertainty estimates by leveraging unlabeled data, for example from a target domain of interest or from data augmentation techniques, an approach that has proven extraordinarily successful in the field of self-supervised and representation learning (Chen et al., 2020; Guo et al., 2022; Caron et al., 2021).

We analyze the practical effectiveness of CSD through an empirical evaluation on a variety of distribution shift detection tasks (Van Amersfoort et al., 2020) using the FashionMNIST, MNIST, KMNIST, and NOTMNIST datasets (Xiao et al., 2017; Deng, 2012; Clanuwat et al., 2018). We moreover use CSD to generate an exploration signal on sparse-reward reinforcement learning problems from the visual RL benchmark VizDOOM (Kempka et al., 2016). Empirically, CSD consistently achieves competitive and sometimes superior uncertainty estimation to finite deep ensembles and other baseline methods while maintaining lower computational cost. We believe these results establish CSD as a both principled and scalable alternative to ensemble-based uncertainty quantification and exploration methods.

## 2 BACKGROUND

For our default framework, we consider a finite Markov Decision Process (MDP, Bellman, 1957) of the tuple $(\mathcal{S}, \mathcal{A}, \mathcal{R}, \gamma, P, \mu)$, with state space $\mathcal{S}$, action space $\mathcal{A}$, immediate reward distribution $\mathcal{R} : \mathcal{S} \times \mathcal{A} \to \mathscr{P}(\mathbb{R})$, discount $\gamma \in [0, 1]$, transition kernel $P : \mathcal{S} \times \mathcal{A} \to \mathscr{P}(\mathcal{S})$, and the start state distribution $\mu \in \mathscr{P}(\mathcal{S})$. Here, $\mathscr{P}(\mathcal{Z})$ indicates the space of probability distributions over some space $\mathcal{Z}$ and random variables are denoted with uppercase letters. Given a state $S_t$ at time $t$, agents choose an action $A_t$ from a stochastic policy $\pi : \mathcal{S} \to \mathscr{P}(\mathcal{A})$ and subsequently receive the immediate reward $R_t \sim \mathcal{R}(\cdot|S_t, A_t)$ and observe next state $S_{t+1} \sim P(\cdot|S_t, A_t)$. The expected discounted sum of future rewards, conditioned on a particular state $s$ and action $a$ is known as the state-action value and is given by $Q^\pi(s, a) = \mathbb{E}_{P,\pi}[\sum_{t=0}^\infty \gamma^t R_t | S_0 = s, A_0 = a]$. This value function adheres to a temporal consistency condition described by the Bellman equation (Bellman, 1957)

$$Q^\pi(s, a) = \mathbb{E}_{P,\pi}[R_0 + \gamma Q^\pi(S_1, A_1)|S_0 = s, A_0 = a], \quad (1)$$

where $\mathbb{E}_{P,\pi}[\cdot]$ indicates that $S_1$ and $A_1$ are drawn from $P$ and $\pi$ respectively. The expected return of a policy $\pi$ can compactly be expressed through the state-action value and the starting state distribution through

$$J(\pi) = \mathbb{E}_{S_0 \sim \mu, A_0 \sim \pi}[Q^\pi(S_0, A_0)]. \quad (2)$$

The objective of reinforcement learning is to find an optimal policy $\pi^*$ that maximizes the above equation $\pi^* = \arg\max J(\pi)$.

### 2.1 EXPLORATION IN REINFORCEMENT LEARNING

A fundamental challenge in attaining an optimal policy $\pi^*$ lies in the exploration-exploitation tradeoff: an agent must decide whether to exploit its current knowledge to maximize returns or whether to explore novel actions in order to discover better strategies. Efficient exploration is particularly crucial

in high-dimensional or sparse-reward settings, where naive strategies such as random exploration require prohibitive amounts of interactions.

A widely used approach to exploration is *optimism in the face of uncertainty* (Auer et al., 2008; Auer, 2002), where agents prioritize actions with high epistemic uncertainty in value estimates. In the context of model-free RL, provably efficient algorithms often rely on the construction of an upper confidence bound (UCB) that overestimates the true optimal value $Q^{\pi^*}(s, a)$ with high probability (Jin et al., 2018; 2020; Neustroev and de Weerdt, 2020). This may be implemented by adding a well-chosen exploration bonus $b(s, a)$ to value estimates according to

$$Q^{\mathrm{opt}}(s, a) = Q^{\pi}(s, a) + b(s, a). \tag{3}$$

In small state-action spaces, such bonuses can be derived from count-based concentration inequalities (Bellemare et al., 2016; Jin et al., 2020), whereas high-dimensional, continuous domains usually require function approximation, significantly complicating efficient uncertainty estimation (Ghavamzadeh et al., 2015; Osband et al., 2016; Lakshminarayanan et al., 2017; Burda et al., 2019).

With the widespread use of deep neural networks, deep ensembles (Lakshminarayanan et al., 2017) based on random initialization have become a dominant tool for quantifying epistemic uncertainty in high-dimensional continuous spaces (Chen et al., 2017; Osband et al., 2019; He et al., 2020). An informal intuition behind the effectiveness of ensembles is the tendency of randomly initialized NNs to converge to diverse minima in the training loss landscape (Fort et al., 2020), leading to higher prediction diversity for unseen inputs. The variance among ensemble members can then be used to measure the model's uncertainty for a specific input.

## 2.2 Neural Tangent Kernel Gaussian Processes

In order to better understand the properties of deep ensembles and to design better exploration algorithms, an analytical description of deep neural networks and their learning dynamics is desirable. While a general framework remains elusive, significant progress has been made in the field of deep learning theory. In particular, seminal works by Jacot et al. (2018) and Lee et al. (2020) have shown that wide neural networks trained by gradient descent are well-described by their linearized training dynamics and thus predictable.

For this, let neural networks be parametrized functions $f(x, \theta_t) : \mathbb{R}^n \to \mathbb{R}$ and denote training data $\mathcal{X} = \{x_i \in \mathbb{R}^n | i \in \{1, ..., N_D\}\}$ and training labels $\mathcal{Y} = \{y_i \in \mathbb{R} | i \in \{1, ..., N_D\}\}$. We assume training is performed using gradient descent with infinitesimal step sizes, also referred to as gradient flow. The initialization weights $\theta_0$ are drawn i.i.d. from a normal distribution $\theta_0 \sim \mathcal{N}$, and deep ensembles are formed by training multiple independently initialized neural network functions. We furthermore assume so-called NTK-parametrization, which scales forward and backward passes in proportion to layer widths (see Jacot et al., 2018; Lee et al., 2020, for details).

A key result by Lee et al. (2020) is that in the limit of infinite layer widths, the training dynamics of deep networks are described exactly by a Taylor expansion around the parameter initialization $\theta_0$. In this setting, the NTK $\Theta(x, x') : \mathbb{R}^{n \times n} \to \mathbb{R}$, first described by Jacot et al. (2018), emerges as the defining function governing learning dynamics:

$$\Theta_0(x, x') = \nabla_\theta f(x, \theta_0)^\top \nabla_\theta f(x', \theta_0). \tag{4}$$

The NTK can be interpreted as a similarity measure between inputs based on gradient representations of the inputs $x$ and $x'$. Crucially, Jacot et al. (2018) find that in the limit of infinite layer width, $\Theta(x, x')$ becomes deterministic despite random weight initialization $\Theta_0(x, x') = \Theta(x, x')$ and remains constant throughout training, inducing analytically solvable training dynamics. As a result, the post-training NN function $f(x, \theta_\infty)$ can be characterized as a deterministic function of the random initialization $f(x, \theta_0)$ through

$$f(x, \theta_\infty) = f(x, \theta_0) + \Theta(x, \mathcal{X})\Theta(\mathcal{X}, \mathcal{X})^{-1}(\mathcal{Y} - f(\mathcal{X}, \theta_0)). \tag{5}$$

Here, we have overloaded notation to indicate the vectorization $\Theta(x, \mathcal{X}) \in \mathbb{R}^{1 \times N_D}$, $\Theta(\mathcal{X}, \mathcal{X}) \in \mathbb{R}^{N_D \times N_D}$, and so forth. The matrix $\Theta(\mathcal{X}, \mathcal{X})$ is also known as the training Gram matrix, as we will refer to it. Further extending this framework, He et al. (2020) demonstrate that by introducing suitable function priors on $f(x, \theta_0)$, akin to the well-known randomized prior functions by Osband et al.

(2019), the post-training function is described by a Gaussian Process (GP, Rasmussen and Williams, 2006):

$$f(\mathcal{X}_t, \theta_\infty) \sim \mathcal{N}\big( \underbrace{\Theta(\mathcal{X}_t, \mathcal{X})\Theta(\mathcal{X}, \mathcal{X})^{-1}\mathcal{Y}}_{\mathbb{E}[f(\mathcal{X}_t, \theta_\infty)]}, \ \underbrace{\Theta(\mathcal{X}_t, \mathcal{X}_t) - \Theta(\mathcal{X}_t, \mathcal{X})\Theta(\mathcal{X}, \mathcal{X})^{-1}\Theta(\mathcal{X}, \mathcal{X}_t)}_{\mathrm{Cov}[f(\mathcal{X}_t, \theta_\infty)]} \big), \quad (6)$$

where $\mathcal{X}_t$ is an arbitrary test data set. An outline of the derivation of Equations 5 and 6 is provided in Appendix A. Consequently, the variance of an ensemble over infinite random initializations is given by

$$\mathbb{V}[f(x, \theta_\infty)] = \Theta(x, x) - \Theta(x, \mathcal{X})\Theta(\mathcal{X}, \mathcal{X})^{-1}\Theta(\mathcal{X}, x). \quad (7)$$

The above expression provides us with a theoretical footing for understanding the behavior and uncertainty estimates of deep ensembles. In the following sections we will describe our approach for estimating Eq. 7 not as the result of training several random models but deterministically with a single model.

## 3 CONTEXTUAL SIMILARITY DISTILLATION

We now proceed to describe our approach, *contextual similarity distillation* (CSD). The main objective of our method is to approximate the variance of an infinite deep ensemble, as described by Eq. 7, directly with a single model.

### 3.1 ENSEMBLE VARIANCE PREDICTIONS FOR A PRIORI QUERIES

We introduce the underlying idea of CSD in the simplified setting of *a priori known* test points. Given a test query point $x_t$, it is our goal to estimate the variance $\mathbb{V}[f(x_t, \theta_\infty)]$ of an ensemble of independently initialized NNs, trained on a dataset $\mathcal{X}$. It is important to note that one could in principle obtain this variance via the NTK GP by solving Eq. 7. This, however, requires inversion of the potentially very large Gram matrix $\Theta(\mathcal{X}, \mathcal{X})$, which becomes computationally prohibitive for most datasets and models of interest, including RL applications where sample sizes can go into the billions.

Instead of solving Eq. 7 directly, we leverage an alternative perspective that arises naturally from the learning dynamics of wide neural networks. Specifically, we begin with the simple observation that the variance of a wide ensemble at a test point $x_t$ can be computed efficiently as the solution to a regular supervised regression problem of a single model with a particular label function. For this, let $g(x, \tilde{\theta}_t)$ be a NN such that its NTK is equal to $f$ with $\Theta_g(x, x') = \Theta(x, x')$ (i.e., with the same architecture and initial weight distribution). Recall that the post-training NN function $g(x, \tilde{\theta}_\infty)$ with squared loss on $\mathcal{Y}$ is given by

$$g(x, \tilde{\theta}_\infty) = g(x, \tilde{\theta}_0) + \Theta_g(x, \mathcal{X})\Theta_g(\mathcal{X}, \mathcal{X})^{-1}(\mathcal{Y} - g(\mathcal{X}, \tilde{\theta}_0)). \quad (8)$$

It is straightforward to see that for small function initialization[1] $g(x, \tilde{\theta}_0) \approx 0$, $\forall x$ the r.h.s. of this expression, when choosing the label function $\mathcal{Y}_{x_t}(\mathcal{X}) = \Theta(\mathcal{X}, x_t)$, simplifies to

$$g_{x_t}(x, \tilde{\theta}_\infty) = \Theta(x, \mathcal{X})\Theta(\mathcal{X}, \mathcal{X})^{-1}\Theta(\mathcal{X}, x_t), \quad (9)$$

where we used the subscript $x_t$ to indicate the function's dependence on the label function $\mathcal{Y}_{x_t}$. This identity now recovers exactly the problematic right term of Eq. 7 containing the Gram inversion $\Theta(\mathcal{X}, \mathcal{X})^{-1}$. Note that $g_{x_t}(x, \tilde{\theta}_\infty)$ is obtained "naturally" as the result of gradient-based regression, without requiring explicit inversion of $\Theta(\mathcal{X}, \mathcal{X})$ or training of a large ensemble at any point. The ensemble variance in a query point $x_t$ can be obtained as

$$\mathbb{V}[f(x_t, \theta_\infty)] = \Theta(x_t, x_t) - g_{x_t}(x_t, \tilde{\theta}_\infty), \quad (10)$$

which can be computed efficiently. Fig. 1 illustrates the above-described process of obtaining expression 10 geometrically. While simple, we believe this formulation provides a crucial insight: uncertainty estimation for a NN can be phrased as a singular prediction problem of kernel similarities.

---

[1]For example, small function initialization can simply be obtained by redefining $\hat{f}(x, \theta_t) := f(x, \theta_t) - f(x, \theta_0)$.

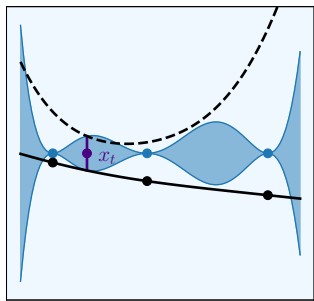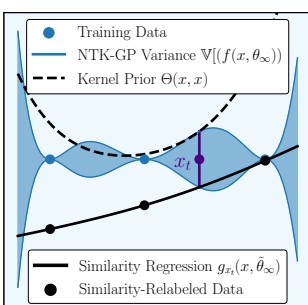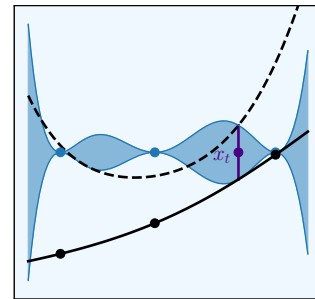

Figure 1: Illustration of regression tasks with query-dependent NTK similarities as labels. The difference between the kernel prior function $\Theta(x, x)$ (dotted line) and the post-training regression function $g_{x_t}(x, \tilde{\theta}_\infty)$ matches exactly ensemble variance in $x_t$ note that we shifted curves in black by a constant offset for each subplot to illustrate this equality). Plots from left to right depict the same principle, but for different query points $x_t$.

## 3.2 Ensemble Variance Estimation for Arbitrary Query Points

In the above derivation, we outlined an efficient method for obtaining ensemble variances at a specific test query point $x_t$ *known a priori*. An obvious limitation of this approach, however, is that the used labeling function $\mathcal{Y}_{x_t}(\mathcal{X}) = \Theta(\mathcal{X}, x_t)$ and by extension the model $g_{x_t}(x, \tilde{\theta}_\infty)$ is inherently dependent on the test point $x_t$ and not usable for arbitrary queries.

To overcome this limitation, we now formulate a *contextualized regression model* $g(x, c, \tilde{\theta}_t)$, where $c$ serves as a context variable that determines the label function used during training of the function $g(x, c, \tilde{\theta}_t)$. Specifically, instead of defining a label function that depends on a single fixed test query $x_t$, we construct a family of label functions parameterized by the context $c$, $\mathcal{Y}_c(\mathcal{X}) = \Theta(\mathcal{X}, c)$. This means that for a set of context data $\mathcal{C} = \{c_i \in \mathbb{R}^n | i \in \{1, ..., N_C\}\}$, the model $g(x, c, \tilde{\theta}_t)$ is optimized to solve a supervised regression problem associated with labels $\mathcal{Y}_c(\mathcal{X})$.

Intuitively, this approach can be interpreted as an attempt to interpolate between multiple regression solutions that were trained on the same dataset $\mathcal{X}$ but with different label functions $\mathcal{Y}_c(\mathcal{X})$. Geometrically, this corresponds to conjoining the functions $g_{x_t}$ in Fig. 1 along a new dimension $c$. So long as $g(x, c, \tilde{\theta}_\infty)$ maintains the approximate dynamics of $g_c(x, \tilde{\theta}_\infty)$, this model can be evaluated quickly for arbitrary test points by setting $c = x_t$ in

$$g(x, c, \tilde{\theta}_\infty) \approx \Theta(x, \mathcal{X})\Theta(\mathcal{X}, \mathcal{X})^{-1}\Theta(\mathcal{X}, c). \tag{11}$$

This generalization accordingly enables ensemble variance estimation across arbitrary points $x$ without requiring a separate regression solution for each individual query by computing

$$\mathbb{V}[f(x, \theta_\infty)] \approx \Theta(x, x) - g(x, x, \tilde{\theta}_\infty). \tag{12}$$

An intuitive interpretation of the function $g(x, x, \tilde{\theta}_\infty)$ is that it captures an ensemble's confidence gained through observing the training data $\mathcal{X}$, weighted by its similarity to $x$. The resulting variance of Eq. 12 can then be understood as the difference between a prior uncertainty term $\Theta(x, x)$ and the confidence term $g(x, x, \tilde{\theta}_\infty)$. One should note at this point, that the evaluation of $g(x, c, \tilde{\theta}_\infty)$ for contexts $c \notin \mathcal{C}$ not used during training requires $g$ to generalize to novel $c$. Furthermore, the introduction of the context variable $c$ may influence the training dynamics of $g$, putting this approach into the realm of approximate algorithms. We have added a section to Appendix B.1 that discusses and summarizes used approximations and their implications for practical settings.

**Finetuning Variance Estimates with Context Data.** Before proceeding to describe our practical setup, we outline a property of contextualized similarity distillation that emerges through the above-described modeling choices. Our theoretical motivation highlights that *exact* ensemble variances (in the NTK regime) can be obtained when the test point $x_t$ is known a priori. The implication of the subsequent formulation as a contextualized regression problem is that, when available, one

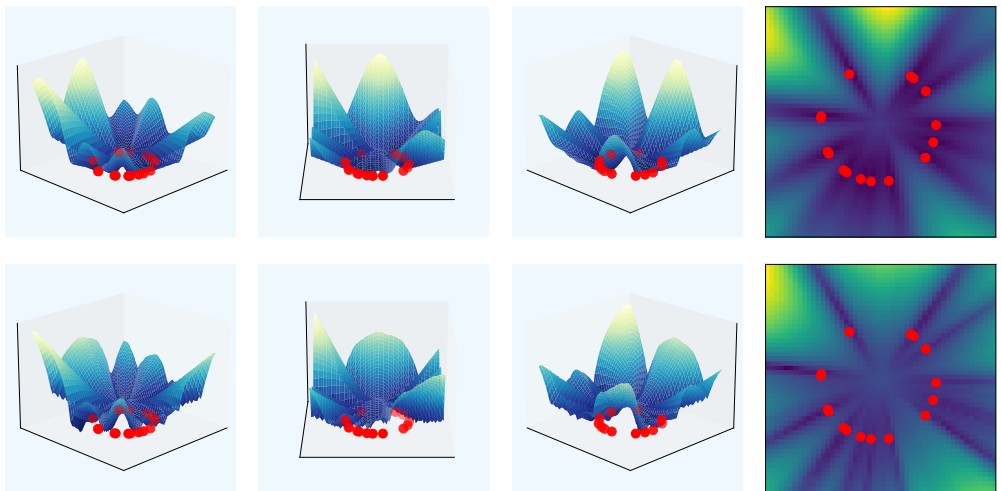

Figure 2: *Top Row*: Variance of an ensemble of 100 randomly initialized neural networks on a 2D toy regression task. Red dots are training points. *Bottom Row*: Variance prediction by contextual similarity distillation (CSD) with a single model on the same regression task.

can include unlabeled context data $\mathcal{C}$ during training to obtain better uncertainty estimates in the domain of interest, as we will show later in the experimental section. This property also opens up the possibility of using unlabeled data augmentations to improve uncertainty estimation, an approach that has proven extraordinarily successful in the field of self-supervised and representation learning (Chen et al., 2020; Guo et al., 2022; Caron et al., 2021) and not easily incorporated with standard approaches for uncertainty estimation (Lakshminarayanan et al., 2017; Gal and Ghahramani, 2016; Burda et al., 2019).

### 3.3 CONTEXTUALIZED SIMILARITY DISTILLATION WITH DEEP NEURAL NETWORKS

Building on this theoretical basis, we proceed to describe a setting for contextualized similarity distillation with deep neural networks. This section outlines algorithmic design choices we found to be computationally efficient while maintaining the approach's theoretical motivation.

First, we parameterize the contextualized regression model $g(x, c, \tilde{\theta}_\infty)$ as an inner product between a feature vector $\phi(x, \tilde{\theta}_{\text{feat}})$ and a context vector $\psi(c, \tilde{\theta}_{\text{ctxt}})$ as

$$g(x, c, \tilde{\theta}_\infty) = \phi(x, \tilde{\theta}_{\text{feat}})^\top \psi(c, \tilde{\theta}_{\text{ctxt}}). \tag{13}$$

Conceptually, this parametrization can be thought of as introducing a context-dependent final layer of weights, represented by $\psi(c, \tilde{\theta}_{\text{ctxt}})$, to the regression model $g$. Computationally, this inner product parametrization bears the advantage that $g(\mathcal{X}, \mathcal{C}, \tilde{\theta}_\infty) \in \mathbb{R}^{N_D \times N_C}$ can be evaluated quickly without requiring explicit forward passes for each pairing $(x_i \in \mathcal{X}, c_j \in \mathcal{C})$.

Second, we approximate the NTK prior $\Theta(x, x')$ with partial gradients. Given that $\Theta(x, x')$ is not involved in backward gradient computations, computing the full analytical or empirical prior kernel functions $\Theta(x, x')$ is often not computationally prohibitive, but can pose a burden for models with large parameter spaces. We find that gradients with respect to only the last layer weights $\theta_0^L$ are sufficient in practice and further accelerate computation. Assuming, the last layer of $f$ is a dense layer such that $f(x, \theta_0) = \varphi(x, \theta_0^{1:L-1})^\top \theta_0^L$, we have

$$\Theta^L(x, x') = \nabla_{\theta_0^L} f(x, \theta_0)^\top \nabla_{\theta_0^L} f(x', \theta_0) = \varphi(x, \theta_0^{1:L-1})^\top \varphi_f(x', \theta_0^{1:L-1}). \tag{14}$$

The resulting training pipeline for $g(x, c, \tilde{\theta}_t)$ involves a simple supervised regression task with minimization of the squared loss, where $(x_i, c_i)$ are sampled randomly from $\mathcal{X}$ and $\mathcal{C}$

$$\mathcal{L}(\tilde{\theta}_t) = \frac{1}{N} \sum_i^N \frac{1}{2} \big(g(x_i, c_i, \tilde{\theta}_t) - \Theta^L(x_i, c_i)\big)^2. \tag{15}$$

Table 1: Distribution Shift Detection. Test accuracy and average OOD detection metrics across MNIST, FashionMNIST, KMNIST, NotMNIST. OOD metrics are evaluated for each ID dataset against the remaining OOD datasets and a perturbed version of the ID dataset.

| Method | Acc. | AUROC | AUPR-IN | AUPR-OUT |
|---|---|---|---|---|
| MCD | $94.39 \pm 0.10$ | $85.67 \pm 0.21$ | $81.73 \pm 0.34$ | $86.44 \pm 0.20$ |
| BNN-MCMC | $87.70 \pm 0.38$ | $83.17 \pm 0.60$ | $82.65 \pm 0.66$ | $82.28 \pm 0.71$ |
| BNN-Laplace | $90.86 \pm 0.62$ | $81.38 \pm 0.73$ | $79.43 \pm 0.84$ | $81.84 \pm 0.66$ |
| RND | $96.18 \pm 0.05$ | $\mathbf{94.40} \pm 0.41$ | $94.17 \pm 0.63$ | $\mathbf{94.01} \pm 0.31$ |
| ENS(3) | $96.91 \pm 0.04$ | $92.30 \pm 0.09$ | $92.83 \pm 0.10$ | $91.37 \pm 0.11$ |
| ENS(15) | $\mathbf{97.18} \pm 0.03$ | $94.00 \pm 0.07$ | $\mathbf{94.70} \pm 0.07$ | $92.99 \pm 0.06$ |
| CSD | $96.29 \pm 0.07$ | $\mathbf{96.63} \pm 0.35$ | $\mathbf{96.94} \pm 0.39$ | $\mathbf{96.19} \pm 0.32$ |
| CSD-Aug. | $96.28 \pm 0.06$ | $\mathbf{98.22} \pm 0.14$ | $\mathbf{98.51} \pm 0.13$ | $\mathbf{97.80} \pm 0.17$ |
| CSD-OOD. | $96.30 \pm 0.06$ | $\mathbf{98.57} \pm 0.14$ | $\mathbf{98.86} \pm 0.12$ | $\mathbf{98.19} \pm 0.15$ |

Lastly, we propose several choices for the context data $\mathcal{C}$. We find that the arguably simplest choice, that is to reuse the training set $c_i \sim \mathcal{X}$, works well in practice and is easily implemented. In addition, it is possible to apply data augmentations to the training samples $\mathcal{X}$ when using as context data. For this, we employ the well-established set of augmentations from the contrastive learning literature (Chen et al., 2020). We note here, that designing novel data augmentation techniques for the purpose of uncertainty quantification is a promising avenue (see for example works by Wen et al. (2020) and Wu and Williamson (2024)). Unlike contrastive learning and many other self-supervised methods, our approach does not require data augmentations to preserve the nature of the original label and can in principle use any unlabeled data. Finally, when available, unlabeled data from the test distribution of interest can be used and often provides an additional improvement in uncertainty estimation, as we will show empirically.

## 4 EMPIRICAL EVALUATION

Our empirical evaluation aims to provide us with a better understanding of contextual similarity distillation in practice. Given that our approach introduces approximations beyond the theoretical framework, we investigate whether CSD maintains its theoretically motivated properties in practice with high-dimensional problem and parameter spaces. Specifically, we aim to assess whether CSD provides a scalable alternative to deep ensembles and other established methods in uncertainty quantification, including Monte Carlo dropout (Gal and Ghahramani, 2016), a Bayesian NN based on Markov chain Monte Carlo sampling (BNN - MCMC, Garriga-Alonso and Fortuin, 2021), a Laplace approximated Bayesian NN (BNN - Laplace, Immer et al., 2021), deep ensembles of sizes 3 and 15 (ENS, Lakshminarayanan et al., 2017) and random network distillation (RND, Burda et al., 2019). Furthermore, we analyze how algorithmic design choices, such as the choice of context data, influence uncertainty estimates. Lastly, we seek to evaluate our approach's efficacy as an exploration signal for deep reinforcement learning agents on sparse-reward visual exploration tasks from the VizDoom (Kempka et al., 2016) suite.

Code for experimental reproduction is available at
`github.com/anyboby/contextual-similarity-distillation` ,
`github.com/anyboby/contextual-similarity-distillation-vizdoom` .

### 4.1 DISTRIBUTION SHIFT DETECTION

Following prior work (Van Amersfoort et al., 2020; Immer et al., 2021; Rudner et al., 2022), we evaluate uncertainty estimates in image classification under distribution shift, where a model trained on an in-distribution dataset is evaluated on inputs from a shifted distribution.

In particular, we train models on one of the FashionMNIST, MNIST, KMNIST, NotMNIST datasets and evaluate uncertainty estimates on the other, shifted datasets and a perturbed version of the in-distribution dataset. Well-calibrated epistemic uncertainty estimates will correlate with dataset shift, such that out-of-distribution samples are likely to be rated more uncertain than in-distribution

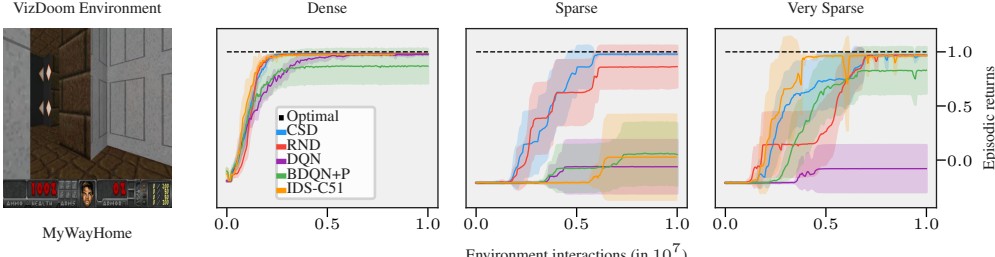

Figure 3: (Left): Visual observation in the VizDoom environment (Kempka et al., 2016). (From Second Left to Right): Mean learning curves in variations of VizDoom *MyWayHome*. Shaded regions are 90% Student's t confidence intervals from 10 seeds.

samples. To compare methods quantitatively, we use the threshold-independent area under the receiver operating characteristic curve (AUROC) metric, as well as the area under the precision-recall curve for in-distribution (AUPR-IN) and out-of-distribution (AUPR-OUT) samples. The AUROC metric can be interpreted as the likelihood of an OOD sample receiving higher uncertainty than an ID sample, while AUPR-IN and AUPR-OUT provide additional sensitivity to dataset size and the choice of the positive class. For these metrics, Table 1 reports the average and standard deviation over 10 seeds, averaged over all permutations of ID and OOD datasets, along with average test accuracy. Full detailed results are provided in the supplementary material.

To analyze the role of the used context data, we evaluate three versions of CSD: a baseline that only uses training data (CSD), a variant incorporating data augmentations to training samples (CSD-Aug.), and a model using context data from the evaluation distribution (CSD-OOD). Even in the basic version, CSD demonstrates highly effective distribution shift detection, surpassing baseline methods on a variety of datasets while requiring only a single model. Our results furthermore suggest that incorporating data augmentations and target-distribution context data indeed significantly improves performance.

## 4.2 EXPLORATION IN VIZDOOM

We now evaluate CSD in a reinforcement learning task with high-dimensional observation spaces and sparse rewards. For this, we consider visual navigation tasks in the VizDOOM environment, where agents explore a 3D maze-like environment with ego-perspective image observations. The agent is tasked with reaching a goal while receiving a minimal constant negative reward except upon successful completion, where a reward of 1 is given. We consider three variations of the task, where agents are initialized at increasing distances from the goal, defining progressively harder exploration tasks (details provided in Appendix C.2).

We use a DQN agent (Mnih et al., 2015) as a base algorithm and include uncertainty estimates by CSD as an intrinsic reward (full details provided in Appendix C). For a comparative evaluation, we compare the performance of CSD-based exploration with several baseline algorithms, including deep Q networks (DQN, Mnih et al., 2015), random network distillation (RND, Burda et al., 2019), bootstrapped Q-networks (BDQN+P, Osband et al., 2019), and information-directed sampling (IDS, Nikolov et al., 2019). Fig. 3 shows mean learning curves across 10 random seeds. Interestingly, the sparse version of the environment appears to be the hardest, a circumstance we believe is due to the spawning point lying in a sidearm of the maze map. Of the tested methods, only CSD was able to find the goal across all seeds and environments, with RND performing most competitively.

## 5 RELATED WORK

Our work builds on the extensive body of literature in the field of uncertainty quantification in deep learning and reinforcement learning. Ensemble learning (Dietterich, 2000) has emerged as on the most effective and reliable approaches to uncertainty estimation (Lakshminarayanan et al., 2017)

and has been widely adopted in the deep reinforcement learning literature. In particular, ensembles can be used for efficient exploration by sampling random models (Osband et al., 2016; Qin et al., 2022; Osband and Van Roy, 2017), by constructing upper confidence bounds for exploration bonuses (Chen et al., 2017; O'Donoghue et al., 2018) or by estimating information gain (Nikolov et al., 2019). Several works moreover rely on deep ensembles to reduce overestimation and improve learning stability (Fujimoto et al., 2018; Haarnoja et al., 2018; Chen et al., 2021), extending to the challenging offline setting (An et al., 2021; Agarwal et al., 2020; Smit et al., 2021).

A number of previous works have focused on reducing ensemble size, notably by disaligning the Jacobian of networks (An et al., 2021), adding repulsive loss terms (Sheikh et al., 2022), or through architectural diversification (Osband et al., 2019; Zanger et al., 2024). Notably, various works aim to quantify epistemic uncertainty with a single model (Pathak et al., 2017; Burda et al., 2019; Filos et al., 2021; Guo et al., 2022; Lahlou et al., 2021), often by measuring prediction errors. To the best of our knowledge, few single-model methods in the field offer an interpretation as ensemble or posterior uncertainty.

In a broader sense, ensembles have been studied extensively from a Bayesian perspective (Hoffmann and Elster, 2021; D'Angelo and Fortuin, 2021). In particular, some of our work relies on the NTK GP characterization of deep ensembles by He et al. (2020), who, in turn, rely on seminal work by seminal work on the NTK by Jacot et al. (2018) and Lee et al. (2020). Subsequent analysis has used the NTK to disentangle ensemble variance (Kobayashi et al., 2022). Recent works Wilson et al. (2025) rely on NTK theory to derive a sampling-based uncertainty estimator, while Calvo-Ordoñez et al. (2024) construct uncertainty estimates using several regression models. In contrast to the latter, our method uses a contextualized regression model that allows for single-model uncertainty estimates in a deep learning setting.

## 6 CONCLUSION

This work introduced *contextual similarity distillation* (CSD), a novel single-model approach for uncertainty quantification that estimates the predictive variance of an ensemble with a single model and forward pass. By reframing ensemble variance estimation as a structured regression problem, CSD enables efficient uncertainty estimation without requiring the training of multiple models, stochastic forward passes, or explicit kernel matrix inversion. Instead, phrasing predictive variance estimation as a contextualized regression problem is amenable to standard training pipelines with deep NNs and gradient descent.

We implemented CSD in a deep learning setting and performed a comparative evaluation on a variety of distribution shift detection and reinforcement learning tasks. Empirically, we found that CSD provides uncertainty estimates competitive and sometimes superior to deep ensembles and other alternatives on all tasks. This makes CSD an attractive option for guiding exploration in RL, as our experiments on high-dimensional exploration tasks confirmed. Our results furthermore confirmed that our approach can leverage unlabeled target domain data and data augmentations to further refine uncertainty estimates. We believe our work opens up several avenues for future research. Due to its conceptual similarity to contrastive learning approaches, we believe refining the generation and incorporation of contextual data through augmentation is an exciting avenue for research that is currently not commonplace in the context of uncertainty quantification. Moreover, a natural extension of our approach could aim to include an explicit quantification of aleatoric uncertainty so as to provide a complete separation of aleatoric and epistemic uncertainties within one model. Although our current derivations to not consider the learning dynamics of such probabilistic models, we believe such an extension to be feasible and valuable. Lastly, our approach could be leveraged to drive exploration in various hard exploration tasks or to drive stability in offline RL.

Our findings, we believe, position CSD as a scalable alternative to deep ensembles, offering a principled and computationally efficient method for uncertainty quantification in deep learning.

## 7 ACKNOWLEDGEMENTS

This project has received funding from the EU Horizon 2020 programme *Epistemic AI* under grant number 964505 and the Dutch Research Council (NWO) project *Reliable Out-of-Distribution Generalization in Deep Reinforcement Learning* with project number OCENW.M.21.234. Computational resources for experimental studies were provided by the Delft High Performance Computing Centre (DHPC) and the Delft Artificial Intelligence Cluster  (DAIC).

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

## A    Linearized Neural Network Learning Dynamics

For completeness, we briefly outline a sketch for how the GP interpretation of wide neural networks governed by NTK dynamics described in Expression 6 can be obtained. This section largely follows the seminal works by Jacot et al. (2018), Lee et al. (2020) and He et al. (2020), to whom we refer readers interested in further details.

We begin by constructing a first-order Taylor expansion of the neural network function $f(x, \theta_0)$ around its initialization parameters $\theta_0$:

$$f_{\text{lin}}(x, \theta_t) = f(x, \theta_0) + \nabla_\theta f(x, \theta_0)^\top (\theta_t - \theta_0). \tag{16}$$

When trained on $\mathcal{X}$ and $\mathcal{Y}$ with the squared error loss $\mathcal{L} = \frac{1}{2}\|f_{\text{lin}}(\mathcal{X}; \theta_t) - \mathcal{Y}\|^2$, gradient flow with a learning rate $\alpha$ induces an evolution of $\theta_t$ according to

$$\frac{\mathrm{d}}{\mathrm{d}t}\theta_t = -\alpha \nabla_\theta \mathcal{L} = -\alpha \nabla_\theta f_{\text{lin}}(\mathcal{X}, \theta_t) \nabla_{f_{\text{lin}}(\mathcal{X}, \theta_t)} \mathcal{L}. \tag{17}$$

In function space, this evolution translates to the expression

$$\frac{\mathrm{d}}{\mathrm{d}t} f_{\text{lin}}(x; \theta_t) = \nabla_\theta f_{\text{lin}}(x, \theta_t)^\top \frac{\mathrm{d}}{\mathrm{d}t}\theta_t = -\alpha \Theta_0(x, \mathcal{X})(f_{\text{lin}}(\mathcal{X}; \theta_t) - \mathcal{Y}), \tag{18}$$

where $\Theta_0(x, x') = \nabla_\theta f(x, \theta_0)^\top \nabla_\theta f(x', \theta_0)$ is the (empirical) tangent kernel of $f_{\text{lin}}(x, \theta_t)$. Since this linearization has constant gradients $\nabla_\theta f(x, \theta_0)$, the resulting differential equation is linear and solvable. For the substitution $v_t = (f_{\text{lin}}(\mathcal{X}; \theta_t) - \mathcal{Y})$, we obtain the training error dynamics $\frac{\mathrm{d}}{\mathrm{d}t} v_t = -\alpha \Theta_0(\mathcal{X}, \mathcal{X}) v_t$ to which an exponential ansatz yields the solution

$$f_{\text{lin}}(\mathcal{X}; \theta_t) - \mathcal{Y} = e^{-\alpha t \Theta_0(\mathcal{X}, \mathcal{X})}(f(\mathcal{X}; \theta_0) - \mathcal{Y}), \tag{19}$$

where the matrix exponential $e^{-\alpha t \Theta_0(\mathcal{X}, \mathcal{X})}$ was used. Plugging Eq. 19 back into Eq. 18, one arrives at the identity

$$\frac{\mathrm{d}}{\mathrm{d}t} f_{\text{lin}}(x; \theta_t) = -\alpha \Theta_0(x, \mathcal{X}) e^{-\alpha t \Theta_0(\mathcal{X}, \mathcal{X})}(f(\mathcal{X}; \theta_0) - \mathcal{Y}). \tag{20}$$

This differential expression is explicit in its terms such that we can obtain a solution by integration through

$$f_{\text{lin}}(x; \theta_t) = f(x, \theta_0) + \int_0^t \frac{\mathrm{d}}{\mathrm{d}t'} f_{\text{lin}}(x, \theta_{t'}) \mathrm{d}t' \tag{21}$$

$$= f(x, \theta_0) + \Theta_0(x, \mathcal{X})\Theta_0(\mathcal{X}, \mathcal{X})^{-1}(e^{-\alpha t \Theta(\mathcal{X}, \mathcal{X})} - I)(f(\mathcal{X}, \theta_0) - \mathcal{Y}), \tag{22}$$

which recovers Eq. 5 for $t \to \infty$. A central result by Jacot et al. (2018) and extended in the linearized setting by Lee et al. (2020) is that, as layer widths of the neural network go to infinity, the NTK $\Theta_0(x, x')$ becomes deterministic and constant and the linear approximation $f_{\text{lin}}(x; \theta_t)$ becomes exact w.r.t. the original function $\lim_{\text{width} \to \infty} f_{\text{lin}}(x; \theta_t) = f(x, \theta_t)$.

Rewriting the (infinite width) post-training test and training functions as an affine transformation of the initialization yields

$$\begin{pmatrix} f(\mathcal{X}_t, \theta_\infty) \\ f(\mathcal{X}, \theta_\infty) \end{pmatrix} = \begin{pmatrix} I & -\Theta(\mathcal{X}_t, \mathcal{X})\Theta(\mathcal{X}, \mathcal{X})^{-1} \\ 0 & 0 \end{pmatrix} \begin{pmatrix} f(\mathcal{X}_t, \theta_0) \\ f(\mathcal{X}, \theta_0) \end{pmatrix} + \begin{pmatrix} \Theta(\mathcal{X}_t, \mathcal{X})\Theta(\mathcal{X}, \mathcal{X})^{-1}\mathcal{Y} \\ \mathcal{Y} \end{pmatrix}. \tag{23}$$

For the earlier described parametrization of $f$, the set of initial predictions is known to follow a multivariate Gaussian distribution (Lee et al., 2018) described by the neural network Gaussian process (NNGP) $f(\mathcal{X}, \theta_0) \sim \mathcal{N}(0, \kappa(\mathcal{X}, \mathcal{X}))$ (and analogously for $\mathcal{X}_t$), where

$$\kappa(\mathcal{X}_t, \mathcal{X}_t) = \mathbb{E}_{\theta_0}\left[ f(\mathcal{X}_t, \theta_0) f(\mathcal{X}_t, \theta_0)^\top \right]. \tag{24}$$

Affine transformations of multivariate Gaussian random variables $X \sim \mathcal{N}(\mu_X, \Sigma_X)$ with $Y = a + BX$ are, in turn, multivariate Gaussian random variables with distribution $Y \sim \mathcal{N}(a +$

$B\mu_X$, $B\Sigma_X B^\top$). We here omit explicit derivations and rearrangements for brevity. As a consequence, Eq. 23 with initialization covariance from Eq. 24 is also described by a multivariate Gaussian with mean and covariance given by

$$\mathbb{E}_{\theta_0}[f(\mathcal{X}_t, \theta_\infty)] = \Theta(\mathcal{X}_t, \mathcal{X})\Theta(\mathcal{X}, \mathcal{X})^{-1}\mathcal{Y},$$

$$\mathrm{Cov}(f(\mathcal{X}_t, \theta_\infty)) = \kappa(\mathcal{X}_t, \mathcal{X}_t) - \Theta(\mathcal{X}_t, \mathcal{X})\Theta(\mathcal{X}, \mathcal{X})^{-1}\kappa(\mathcal{X}, \mathcal{X})\Theta(\mathcal{X}, \mathcal{X})^{-1}\Theta(\mathcal{X}, \mathcal{X}_t) \quad (25)$$

$$- \left(\Theta(\mathcal{X}_t, \mathcal{X})\Theta(\mathcal{X}, \mathcal{X})^{-1}\kappa(\mathcal{X}, \mathcal{X}_t) + \mathrm{h.c.}\right),$$

where h.c. refers to the Hermitian conjugate of the preceding term. He et al. (2020) then introduce constant "correction" terms to the function initialization described in Eq. 24, in particular such that $\kappa(x, x') = \Theta(x, x')$. This simplifies Expression 25 significantly and now permits a Gaussian process interpretation with the final expression given by Eq. 6.

## B  Further Discussions

Below, we further discuss the approximate nature of our method and provide a more general discussion of the terminology used in the context of this paper and the broader field of uncertainty quantification.

### B.1  Discussion on Approximations

As our method relies on several approximations, we include a discussion that aims to provide an overview of the approximate nature of our method and in which settings it is exact or where deviations may be more likely.

The first central approximation we make is to model neural networks with dynamics governed by a deterministic and constant NTK. Jacot et al. (2018) show that this is the case for fully connected NNs with NTK parametrization trained on a squared loss. The implied dynamics are solved assuming gradient flow, that is with infinitesimal step sizes and full-batch gradients. Jacot et al. (2018) and Lee et al. (2020) moreover show that convergence and final generalization behavior is empirically well-described by wide but finite architectures including fully connected NNs, convolutional NNs and residual architectures, trained with stochastic gradient descent. The function initialization scheme proposed by He et al. (2020) allows for a Gaussian process interpretation of NNs from random initialization and largely relies on the same assumptions as the above-described works.

Our theoretical motivation, outlined in Sections 3.1 and 3.2, relies on the GP description of deep ensembles and the implied assumptions. Given this setting, that is assuming NTK parametrization with infinite widths, function initialization according to He et al. (2020), and gradient flow with squared loss, the derivation for single-query ensemble variances in Section 3.1 is exact. In our contextualized model described in Section 3.2, we introduce an additional approximation through the introduction of an explicit context variable $c$, which may interfere with the training dynamics of $g(x, c, \tilde{\theta})$. Let training tuples be $x^c = (x, c)$ and $\mathcal{X}^c = \{x_1^c, x_2^c, ..., x_{N_T}^c\}$ and let the NTK of $g$ be $\Theta_g((x, c), (x', c')) = \nabla_{\tilde{\theta}} g(x, c, \tilde{\theta}_0)^\top \nabla_{\tilde{\theta}} g(x', c', \tilde{\theta}_0)$. The analogous regression solution to the function $g(x, c, \tilde{\theta})$ by minimizing the loss in Eq. 15 becomes

$$g(x, c, \tilde{\theta}_\infty) = \Theta_g(x^c, \mathcal{X}^c)\Theta_g(\mathcal{X}^c, \mathcal{X}^c)^{-1}\Theta(\mathcal{X}^c). \quad (26)$$

A natural setting in which these training dynamics recover Eq. 11 is when gradients are independent between context, that is $\Theta_g((x, c), (x, c')) = 0$ if $c \neq c'$ and maintain the gradient structure of $\Theta(x, x')$ with $\Theta_g((x, c), (x', c)) = \Theta(x, x')$, $\forall c \in \mathcal{C}$. However, this setting would hardly permit meaningful interpolations and extrapolations between different contexts $c$, such that one engages in a trade off between generalization capability towards general contexts $c$ and interference in the training dynamics.

Beyond this, our practical setting approximates the NTK prior function with partial gradients as outlined in Eq. 14 of Section 3.3. The influence of this approximation choice generally depends on architecture, but we found it to perform well in our experiments using deep convolutional and residual architectures. Lastly, the RL exploration setting involves data streams rather than fixed datasets $\mathcal{X}$, further deviating from the earlier delineated dynamics. Understanding the influence of this non stationarity on training dynamics is an open problem, and we believe countermeasures like periodic resets (D'Oro et al., 2023) are a promising avenue for future research.

While alleviating and quantifying the assumption stated above are largely open problems in deep learning theory (Hanin and Nica, 2019; Seleznova and Kutyniok, 2022; Yang and Hu, 2021; Cohen et al., 2021; Lewkowycz et al., 2020), various approaches exist that aim to quantify errors w.r.t. more realistic NN behavior, e.g. in the finite-width regime or with discrete gradient descent. We outline one such direction, following work by Lee et al. (2020), to quantify approximation errors of the linearized NN dynamics by assuming

1. NNs of depth $L$ to have equivalent layer-widths $n_1 = n_2 = \ldots = n_L = n$
2. a full-rank analytical NTK, i.e. $\lambda_{\min}(\Theta(\mathcal{X}^c, \mathcal{X}^c)) > 0$, $\lambda_{\max}(\Theta(\mathcal{X}^c, \mathcal{X}^c)) < \infty$
3. a maximum learning rate of $\alpha_0 \leq \alpha_{\text{crit}} = \frac{2}{\lambda_{\min} + \lambda_{\max}}$
4. the contextualized training set $\mathcal{X}^c$ is contained in the unit ball, i.e., $\|x^c\|_2 \leq 1$ for all $x^c \in \mathcal{X}^c$, with distinct elements.
5. nonlinearities $\sigma(x)$ to satisfy

$$|\sigma(0)|, \quad \|\sigma'\|_\infty, \quad \sup_{x \neq x'} \frac{|\sigma'(x) - \sigma'(x')|}{|x - x'|} < \infty \tag{27}$$

6. block diagonality of the contextualized NTK as stated in Eq. (26) such that the infinite-width limit recovers $g(x, c, \tilde{\theta}_\infty) = \Theta(x, \mathcal{X}^c)\Theta(\mathcal{X}^c, \mathcal{X}^c)^{-1}\Theta(\mathcal{X}^c, c)$.

Under these conditions, we have that the infinite-width NN $g(x, c, \tilde{\theta}_\infty)$ behaves as $g(x, c, \tilde{\theta}_\infty) = \Theta(x, \mathcal{X}^c)\Theta(\mathcal{X}^c, \mathcal{X}^c)^{-1}\Theta(\mathcal{X}^c, c)$ and one can show that (Lee et al., 2020), with high probability over random initialization, a linearized NN $g^{\text{lin}}(x, c, \tilde{\theta}_\infty)$ solution approximates a finite-width NN $g^{\text{real}}(x, c, \tilde{\theta}_\infty)$ trained with gradient descent with (non-critical) step size $\alpha_0$ with $\|g^{\text{lin}}(x, c, \tilde{\theta}_\infty) - g^{\text{real}}(x, c, \tilde{\theta}_\infty)\|_2 = \mathcal{O}(\frac{1}{\sqrt{n}})$. Daniely et al. (2016) furthermore show that the empirical tangent kernel at initialization $\Theta_0$ concentrates at the same $\frac{1}{\sqrt{n}}$ rate, such that we have $\|\Theta_0(x^c, x^c) - \Theta(x^c, x^c)\|_2 = \mathcal{O}(\frac{1}{\sqrt{n}})$. Taken together, we can conclude that the finite-width approximation error as occurring in Eq. (12), under the stated conditions, scales as

$$\|\mathbb{V}(f(x, \theta_\infty)) - \big(\Theta_0(x, x) - g^{\text{real}}(x, x, \tilde{\theta}_\infty)\big)\|_2 = \mathcal{O}(\frac{1}{\sqrt{n}}). \tag{28}$$

We refer interested readers to Lee et al. (2020), Daniely et al. (2016), and Hanin and Nica (2019) for detailed derivations and further results going beyond strict NTK-regimes.

## B.2 Discussion on Terminology

While there is broad agreement that it is important to distinguish different sources of uncertainty in machine learning, there remains debate about how these notions should be formally captured and which terminology is appropriate in different contexts. A widely used conceptual distinction is between *epistemic uncertainty* — uncertainty arising from limited knowledge about the true or optimal model parameters — and *aleatoric uncertainty* — uncertainty inherent to the stochasticity in the data-generating process (Hüllermeier and Waegeman, 2020). Together, these sources of uncertainty are considered to constitute *total uncertainty*, though alternative frameworks exist that depart from such an additive decomposition (Shafer, 1976; Cuzzolin, 2021). Several mathematical frameworks aim to formalize these notions: Bayesian inference, arguably the most prominent, capture epistemic uncertainty in the form a posterior distribution over plausible models, given a prior distribution (Neal, 1996). However, alternative frameworks (e.g., frequentist statistics (Le Cam, 2012) or imprecise probability (Shafer, 1976; Walley, 1991; Caprio et al., 2024)) provide alternative perspectives, in part to alleviate the often restrictive requirement of well-specified priors. Our work is situated within the Bayesian viewpoint in the sense that we use the term *epistemic uncertainty* to refer to variability induced by the posterior distribution over functions compatible with the observed data. Within this framework, Gaussian processes offer a nonparametric model class that enables analytical Bayesian inference, and we use the variance of the GP posterior predictive distribution as our measure of epistemic uncertainty. It is important to note, however, that for probabilistic models that explicitly model observation noise (i.e., aleatoric uncertainty), the posterior predictive variance conflates epistemic and aleatoric components and therefore reflects *total* predictive uncertainty rather

than epistemic uncertainty alone (Hüllermeier and Waegeman, 2020). In our setting, however, we assume a deterministic GP model without an observation-noise term, and thus interpret the posterior predictive variance as epistemic uncertainty.

## C  EXPERIMENTAL DETAILS

In the following, we outline details on our experimental setup. This includes hyperparameter settings, hyperparameter search procedures, algorithmic and experimental details, and dataprocessing details.

### C.1  HYPERPARAMETER SETTINGS

In order to facilitate comparable results, our experiments are conducted using a central codebase and follow similar modeling choices such as architectures, optimizer, etc. where sensible. All experiments use a resnet-based model (He et al., 2016) following the IMPALA architecture by Espeholt et al. (2018). We optimized essential and algorithm-specific hyperparameters through a search on a selected subset of experiments.

**Distribution shift detection.**  In the supervised distribution shift detection, we use the IMPALA architecture with $2$ residual blocks and channels widths $32$ and $64$. Hyperparameters were searched on the FashionMNIST dataset as the in-distribution set and the remaining datasets as out-of-distribution sets. Each dataset is normalized to zero-mean and standard deviation $1$ using the training set statistics. For the main classifier we apply random horizontal flips (p=0.5), random vertical flips (p=0.5) and random sized crops (zoom range between 1.0 and 1.3) to training data in all experiments. Learning rate and algorithm-specific hyperparameters were optimized independently, meaning we first performed a search for learning rates, which we used in the (if applicable) subsequent algorithm-specific parameter searches. Table 2 contains lists of all searched parameters, with parenthesis indicating algorithm-specific parameters and italics indicating the parameter used during the learning rate search. The final hyperparameters were chosen based on the average AUROC metric and are reported in Table 4.

**VizDoom.**  In the RL experiments, we conducted a full grid search on the *MyWayHomeSparse* variation of the environment and chose parameters based on performance after $5 \cdot 10^6$ steps. Our basic network architecture is based on the rainbow (Hessel et al., 2018) network proposed by Schmidt and Schmied (2021) who in turn base their architecture on IMPALA (Espeholt et al., 2018) (see also Fig. 4). We use 3 residual blocks with channel widths according to Table 6. Detailed final hyperparameter settings are given in Table 5. We use the same network architecture for value functions and RND/CSD networks (up to output dimensions). All agents furthermore use a data preprocessing pipeline as outlined in Table 6.

### C.2  IMPLEMENTATION DETAILS

In this section, we briefly outline implementation details concerning CSD and the tested baselines.

**Data augmentations**  For both the distribution shift detection experiments (CSD-Aug.) and the VizDoom experiments, we add data augmentation to obtain additional context variables in CSD. In both experiments, we apply augmentations with a probability of $p = 0.25$ and specific augmentations are listed in Table 7.

**Data and context sampling.**  To compute the loss 15, we sample minibatches $\mathcal{X}_{mb}$ from a buffer or data set. Context minibatches $\mathcal{C}_{mb}$ either simply reuse $\mathcal{X}_{mb}$, are generated by applying data augmentations as outlines above, or by sampling from a context data set. We compute inner products over all pairings of the two batches with $\phi(\mathcal{X}_{mb}, \tilde{\theta}_{\text{feat}})^\top \psi(\mathcal{C}_{mb}, \tilde{\theta}_{\text{ctxt}}) \in \mathbb{R}^{N_{mb} \times N_{mb}}$ and compute loss 15 elementwise. Finally, we sum the average diagonal loss and the average off-diagonal loss.

**Normalization.**  During training, we normalize prior features $\bar{\varphi}(x, \theta_0^{1:L-1}) = \varphi(x, \theta_0^{1:L-1})/\|\varphi(x, \theta_0^{1:L-1})\|_2$, feature vectors $\bar{\phi}(x, \tilde{\theta}_{\text{feat}}) = \phi(x, \tilde{\theta}_{\text{feat}})/\|\phi(x, \tilde{\theta}_{\text{feat}})\|_2$, and context vectors $\bar{\psi}(c, \tilde{\theta}_{\text{ctxt}}) = \psi(c, \tilde{\theta}_{\text{ctxt}})/\|\psi(c, \tilde{\theta}_{\text{ctxt}})\|_2$. When computing predictive variances at

Table 2: Searched hyperparameters for distribution shift experiments.

| Hyperparameter | Values |
|---|---|
| Learning rate (All) | $[10^{-4}, 3 \cdot 10^{-4}, 10^{-3}, 3 \cdot 10^{-3}, 10^{-2}, 3 \cdot 10^{-2}, 10^{-1}]$ |
| Dropout probability (MCD) | $[0.05, 0.1, \mathit{0.15}, 0.25, 0.5]$ |
| RND Learning rate (RND) | $[10^{-4}, 3 \cdot 10^{-4}, \mathit{10^{-3}}, 3 \cdot 10^{-3}, 10^{-2}, 3 \cdot 10^{-2}, 10^{-1}]$ |
| CSD Learning rate (CSD) | $[10^{-4}, 3 \cdot 10^{-4}, \mathit{10^{-3}}, 3 \cdot 10^{-3}, 10^{-2}, 3 \cdot 10^{-2}, 10^{-1}]$ |

Table 3: Searched hyperparameters for VizDoom

| Hyperparameter | Values |
|---|---|
| Learning rate (all) | $[1.25 \cdot 10^{-4}, 2.5 \cdot 10^{-4}, 3.75 \cdot 10^{-4}, 5 \cdot 10^{-4}, 6.25 \cdot 10^{-4}, 7.5 \cdot 10^{-4}]$ |
| Loss (all) | [Huber, C51] |
| Prior function scale (BDQN+P, IDS) | [1.0, 3.0, 5.0] |
| Initial bonus $\beta$ (RND, CSD) | [0.05, 0.1, 0.5, 1.0, 5.0, 10.0] |
| RND Learning rate (RND) | $[1.25 \cdot 10^{-4}, 2.5 \cdot 10^{-4}, 3.75 \cdot 10^{-4}, 5 \cdot 10^{-4}, 6.25 \cdot 10^{-4}, 7.5 \cdot 10^{-4}]$ |
| CSD Learning rate (CSD) | $[1.25 \cdot 10^{-4}, 2.5 \cdot 10^{-4}, 3.75 \cdot 10^{-4}, 5 \cdot 10^{-4}, 6.25 \cdot 10^{-4}, 7.5 \cdot 10^{-4}]$ |

Table 4: Hyperparameter settings for distribution shift experiments.

| Hyperparameter | MCMC | Laplace | MCD | ENS | RND | CSD |
|---|---|---|---|---|---|---|
| Main Classifier Network | | | | | | |
| Learning rate | $10^{-3}$ | $10^{-3}$ | $3 \cdot 10^{-4}$ | $10^{-3}$ | $10^{-3}$ | $10^{-3}$ |
| MLP hidden layers | | | 2 | | | |
| MLP layer width | | | 256 | | | |
| Channel Widths | | | 32, 64 | | | |
| RND/CSD Network | | | | | | |
| Learning rate | | | n/a | | $3 \cdot 10^{-3}$ | $10^{-2}$ |
| MLP hidden layers | | | n/a | | 2 | 2 |
| MLP layer width | | | n/a | | 256 | 256 |
| Channel Widths | | | n/a | | 16 | 32 |
| Target hidden layers | | | n/a | | 1 | 1 |
| Output dimensions | | | n/a | | 256 | 256 |
| Ensemble size | n/a | n/a | n/a | 3, 15 | n/a | n/a |
| Dropout rate | n/a | n/a | 0.1 | | n/a | |
| Prior Precision | n/a | 100 | n/a | | n/a | |
| Posterior Temperature | 1.0 | 1.0 | n/a | | n/a | |
| Posterior Samples | 30 | 30 | 100 | | n/a | |
| Epochs per sample | 2 | n/a | n/a | | n/a | |
| Burn-In Epochs | 10 | n/a | n/a | | n/a | |
| Adam epsilon | n/a | $10^{-5}$ | $10^{-5}$ | | $10^{-5}$ | |
| Learning rate anneal | | | Linear | | | |
| Batch size | | | 256 | | | |
| Initialization | | | Orthogonal (Saxe et al., 2013) | | | |

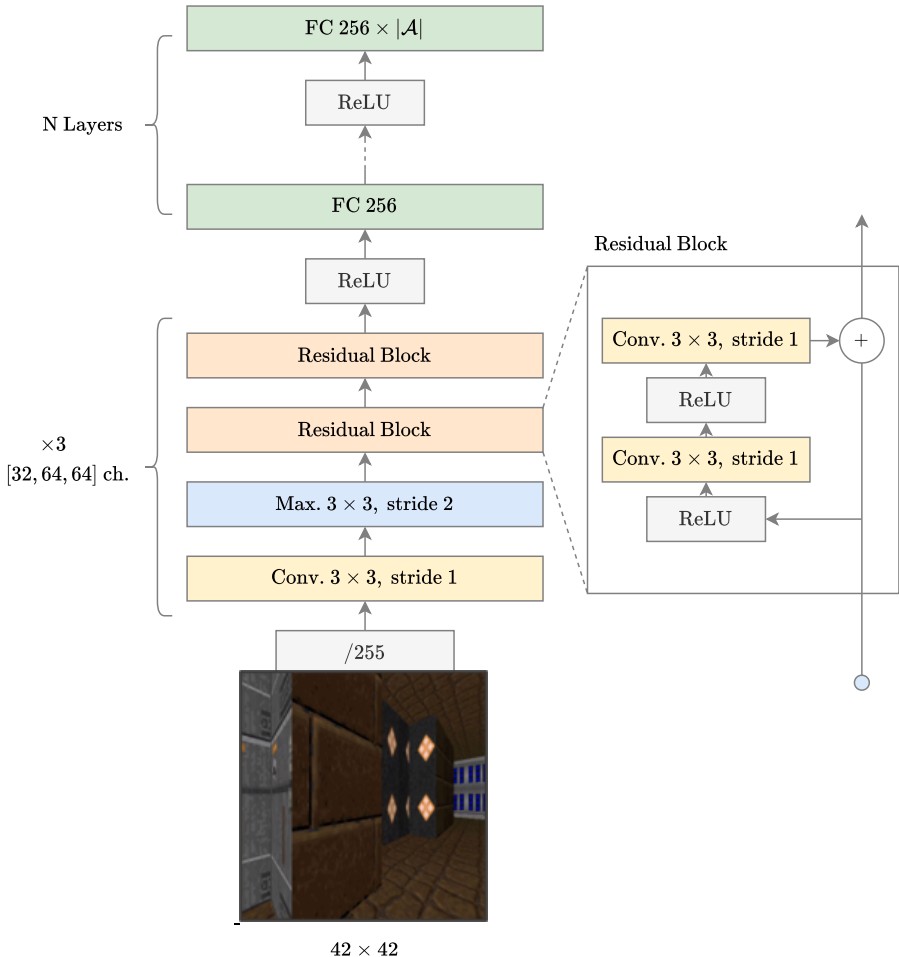

Figure 4: Illustration of the architecture for VizDoom environments. Based on the architecture used by Espeholt et al. (2018).

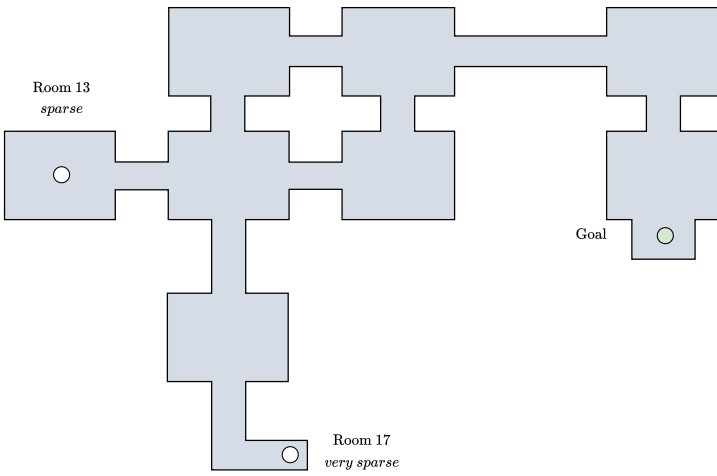

Figure 5: Map for the VizDoom *MyWayHome* environment. Agents are spawned in the *sparse* and *very sparse* locations to vary the exploration difficulty.

Table 5: Hyperparameter settings for VizDoom experiments.

| Hyperparameter | DQN | BDQN+P | RND | IDS | CSD |
|---|---|---|---|---|---|
| Adam Learning rate | $2.5 \cdot 10^{-4}$ | $2.5 \cdot 10^{-4}$ | $6.25 \cdot 10^{-4}$ | $2.5 \cdot 10^{-4}$ | $6.25 \cdot 10^{-4}$ |
| Prior function scale | n/a | 1.0 | n/a | 1.0 | n/a |
| Heads $K$ | 1 | 1 | 101 | 1 / 101 | 101/101 |
| Ensemble size | n/a | 10 | n/a | 10/1 | n/a |
| Initial bonus $\beta_{\text{init}}$ | n/a | n/a | 1.0 | 0.1 | 0.1 |
| Final bonus $\beta_{\text{final}}$ | n/a | n/a | 0.01 | 0.01 | 0.01 |
| Bonus decay frames | n/a | n/a | $3.3 \cdot 10^6$ | $3.3 \cdot 10^6$ | $3.3 \cdot 10^6$ |
| Loss function | Huber | Huber | C51 | Huber/C51 | C51 |
| Channel Widths | | | 32, 32, 64 | | |
| MLP hidden layers | | | 1 | | |
| MLP layer width | | | 256 | | |
| RND / CSD Network Parameters | | | | | |
| Adam Learning rate | n/a | n/a | $2.5 \cdot 10^{-4}$ | n/a | $2.5 \cdot 10^{-4}$ |
| Channel Widths | n/a | n/a | 16, 16, 32 | n/a | 16, 16, 32 |
| MLP hidden layers (main) | n/a | n/a | 3 | n/a | 3 |
| MLP hidden layers (context) | n/a | n/a | n/a | n/a | 1 |
| MLP hidden layers (prior) | n/a | n/a | 1 | n/a | 1 |
| MLP layer width | n/a | n/a | 256 | n/a | 256 |
| Target hidden layers | n/a | n/a | 1 | n/a | 1 |
| Output dimensions | n/a | n/a | 256 | n/a | 256 |
| Initial $\epsilon$ in $\epsilon$-greedy | | | 1.0 | | |
| Final $\epsilon$ in $\epsilon$-greedy | | | 0.01 | | |
| $\epsilon$ decay frames | | | $500,000$ | | |
| Training starts | | | $100,000$ | | |
| Discount | | | 0.997 | | |
| Buffer size | | | $1,000,000$ | | |
| Batch size | | | 256 | | |
| Parallel Envs | | | 16 | | |
| Adam epsilon | | | 0.005/batch size | | |
| Initialization | | | He uniform (He et al., 2015) | | |
| Gradient clip norm | | | 10 | | |
| Regularization | | | spectral normalization (Gogianu et al., 2021) | | |
| Double DQN | | | Yes (Hasselt, 2010) | | |
| Update frequency | | | 1 | | |
| Target lambda | | | 1.0 | | |
| Target frequency | | | 8000 | | |
| PER $\beta_0$ | | | 0.45 (Schaul et al., 2016) | | |
| n-step returns | | | 10 | | |

Table 6: VizDoom Preprocessing

| Parameter | Value |
|---|---|
| Grayscale | Yes |
| Frame-skipping | No |
| Frame-stacking | 6 |
| Resolution | $42 \times 42$ |
| Max. Episode Length | 2100 |

Table 7: Data augmentations for context data.

| Distribution Shift | VizDoom |
|---|---|
| RandomHorizontalFlip($p = 0.25$) | RandomPerspective($p = 0.5$) |
| RandomVerticallFlip($p = 0.25$) | RandomHorizontalFlip($p = 0.5$) |
| Rotate($p = 0.25$) | RandomResizedCrop(r $= [0.75, 1.0]$) |
| GaussianBlur($\sigma = 1.0$, $p = 0.25$) | |
| RandomResizedCrop(r $= [0.75, 1.0]$) | |
| RandomBrightness(r $= [-1.0, 1.0]$, $p = 0.5$) | |
| RandomContrast(r $= [-1.0, 1.0]$, $p = 0.5$) | |

inference time, we rescale by

$$\mathbb{V}[f(x, \theta_\infty)] \approx \|\varphi(x, \theta_0^{1:L-1})\|_2^2 \big( \bar{\varphi}(x, \theta_0^{1:L-1})^\top \bar{\varphi}(x, \theta_0^{1:L-1}) - \bar{\phi}(x, \tilde{\theta}_{\text{feat}})^\top \bar{\psi}(c, \tilde{\theta}_{\text{ctxt}}) \big), \quad (29)$$

to obtain predictions in the original scale again.

**Small function initialization.**    While our theoretical suggests using small function initialization with $g(x, \tilde{\theta}_0) \approx 0$, $\forall x$, preliminary experiments with a reparametrization $\hat{g}(x, \tilde{\theta}_t) := g(x, \tilde{\theta}_t) - g(x, \tilde{\theta}_0)$ showed no significant differences. In our main implementation we thus refrain from using this reparametrization in favor of simplicity.

**Environment Details.**    We conduct experiments on three variations of the VizDoom VizDoom environment *MyWayHome*. A top-down view of environment map is shown in Fig. 5. In the *dense* setting, at the beginning of each episode agents are spawned in random positions of the map, such that the goal position is encountered stochastically without requiring coordinated exploration. The *sparsity* of the problem is increased by changing the agents spawning location deterministically to a room further from the goal position, that is *Room 13* for the *sparse* setting and *Room 17* for the *very sparse* setting. As described in Section 4, the reward function is sparse. A constant reward of $-1 * 10^{-4}$ is given every timestep and a reward of 1 is given for reaching the goal. Episodes are limited to a length of 2100 timesteps.

**Reinforcement Learning Implementation.**    We outline the basic implementation of our DQN-based RL agent. The agent follows the same algorithmic flow as the established DQN-algorithm (Mnih et al., 2015) and subsequent variations (Hessel et al., 2018; Schmidt and Schmied, 2021). The agent maintains a replay buffer of transitions, from which we sample minibatches of transition $\mathcal{X}_{mb} = \{s_i, a_i, r_i, s_i', T_i\}_{i=1}^{N_{mb}}$, where $T_i$ are terminations. $Q$-networks are then updated at a fixed frequency using the sampled minibatch. As is established, we use target networks with slow-moving parameters for value learning.

We provide intrinsic rewards as generated by CSD to the DQN agent to incentivize exploration. For all our experiments including intrinsic rewards (CSD and RND), we use separate value functions for the intrinsic reward and employ *intrinsic reward priors*, a mechanism suggested by Zanger et al. (2024) which includes intrinsic rewards to the forward pass of the value network. This addresses a common issue with intrinsic reward learning as described previously by Rashid et al. (2020) by preventing underestimation of unseen actions. Specifically, intrinsic reward priors redefine the forward pass of the intrinsic $Q$-function according to

$$\hat{Q}_{\text{in}}(s, a, \theta, \theta_{\text{in}}) = Q_{\text{in}}(s, a, \theta) + \tfrac{1}{2} r_{\text{in}}(s, a, \theta_{\text{in}}),$$

where $r_{\text{in}}(s, a, \theta_{\text{in}})$ denotes an intrinsic reward term, in our experiments generated by either RND or CSD with parameters $\theta_{\text{in}}$. The altered function $\hat{Q}_{\text{in}}(s, a, \theta, \theta_{\text{in}})$ is then used as a drop-in replacement for the $Q$-function in the used algorithm.

**Pseudocode for Reinforcement Learning Experiments.**    We provide pseudocode for a DQN agent with CSD in Algorithm 1. For clarity, we omit standard algorithmic details such as double $Q$-learning, distributional value functions, prioritized experience replay, separate value functions for intrinsic reward, and intrinsic reward priors.

---

**Algorithm 1** CSD-DQN

---

1: initialize CSD model $g(s, a, s_c, a_c, \tilde{\theta}_t) = \phi(s, a, \tilde{\theta}_t)^\top \psi(s_c, a_c, \tilde{\theta}_t)$ with $\tilde{\theta}_0$.
2: initialize CSD prior $\Theta^L(s, a, s_c, a_c, c, \theta_p) = \varphi(s, a, \theta_p)^\top \varphi(s_c, a_c, \theta_p)$ with $\tilde{\theta}_p$.
3: initialize $Q$-function $Q(s, a, \theta_t)$ with $\theta_0$ and target parameters $\bar{\theta}_0$.
4: sample initial state $s_0$ from the environment.
5: **for** $t = 1, \ldots, T$ **do**
6:     take action $a \leftarrow \arg\max_{a' \in \mathcal{A}}\{Q(s, a')\}$
7:     obtain observations $(s_t, r_t, T_t)$ from the environment.
8:     store samples $(s_{t-1}, a_{t-1}, r_t, s_t, T_t)$.
9:     sample transition tuple $\{s_i, a_i, r_i, s'_i, T_i\}_{i=1}^{N_{mb}} \sim \mathcal{B}$ from buffer
10:     sample context tuple $\{\hat{s}_i, \hat{a}_i, \hat{r}_i, \hat{s}'_i, \hat{T}_i\}_{i=1}^{N_{mb}} \sim \mathcal{B}$ from buffer
11:     generate intrinsic reward $r_{\text{in}} := \Theta^L(s_i, a_i, s_i, a_i, \tilde{\theta}_p) - g(s_i, a_i, s_i, a_i, \tilde{\theta}_t)$.
12:     generate next action $a'_i := \arg\max_{a' \in \mathcal{A}}\{Q(s'_i, a', \theta_t)\}$.
13:     update $Q$-function $\theta_t \leftarrow \theta_t - \nabla_{\theta_t}\mathcal{L}(\theta_t)$ with

$$\mathcal{L}(\theta_t) = \frac{1}{2N_{mb}} \sum_i^{N_{mb}} \left(r_i + \beta\, r_{\text{in}} + Q(s_i, a_i, \bar{\theta}_t) - Q(s'_i, a'_i, \theta_t)\right)^2.$$

14:     update CSD model $\tilde{\theta}_t \leftarrow \tilde{\theta}_t - \nabla_{\tilde{\theta}_t}\mathcal{L}(\tilde{\theta}_t)$ with

$$\mathcal{L}(\tilde{\theta}_t) = \frac{1}{2N_{mb}} \sum_i^{N_{mb}} \left(g(s_i, a_i, \hat{s}_i, \hat{a}_i, \tilde{\theta}_t) - \Theta^L(s_i, a_i, \hat{s}_i, \hat{a}_i, \tilde{\theta}_p)\right)^2.$$

15:     **if** $t \% \text{freq} == 0$ **then**
16:         update target parameters $\bar{\theta}_t \leftarrow \lambda\theta_t + (1 - \lambda)\bar{\theta}_t$
17:     **end if**
18: **end for**

---

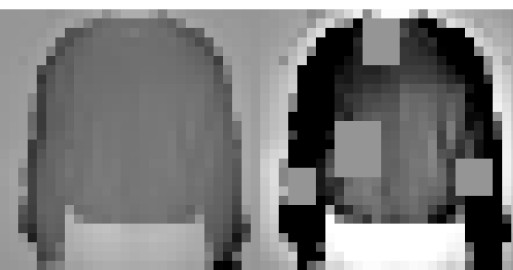

Figure 6: Left: Original Image. Right: Perturbed OOD Image.

### C.3 ADDITIONAL EXPERIMENTAL RESULTS

We report the detailed results of our distribution shift detection tasks. Tables 8 to 11 show OOD detection metrics for the datasets FashionMNIST, MNIST, NotMNIST, and KMNIST. Each table shows the test accuracy and average AUROC, AUPR-IN and AUPR-OUT scores against the remaining three training datasets and an additional perturbed dataset. The perturbed dataset is constructed by applying data augmentations to the ID dataset. In our experiments, we use random brightness changes ($p = 1.0, r = [-1.0, 1.0]$), random contrast changes($p = 1.0, r = [-1.0, 1.0]$), and randomly set patches of an image to zero ($p = 1.0, r = [-1.0, 1.0]$). Fig. 6 shows an example of this.

**Runtime analysis** We report runtime comparisons for all algorithms used in the distribution shift detection tasks in Fig. 7. All *efficient* uncertainty estimation methods, including ours, run faster than the smallest ensemble tested (ENS(3)), while incurring some overhead compared to a single-model baseline (ENS(1)), which does not come with built-in uncertainty quantification capability. Among these, CSD has slightly higher runtime, which is consistent with the algorithmic structure of our method that comprises a separate feature and context model. We note that this runtime comparison is not entirely one-to-one: different methods vary in their application mechanism, with Laplace and MCD being mainly post-hoc methods, while others devise dedicated learning algorithms.

Table 8: Distribution Shift Detection. FashionMNIST as ID dataset.

| Method | Acc. | AUROC | AUPR-IN | AUPR-OUT |
|---|---|---|---|---|
| MCD | $89.24 \pm 0.36$ | $82.23 \pm 0.48$ | $79.88 \pm 0.75$ | $83.01 \pm 0.34$ |
| BNN-MCMC | $85.73 \pm 0.24$ | $85.01 \pm 0.62$ | $85.16 \pm 0.68$ | $83.38 \pm 0.62$ |
| BNN-Laplace | $88.57 \pm 0.80$ | $86.50 \pm 0.67$ | $86.32 \pm 0.75$ | $85.95 \pm 0.75$ |
| RND | $91.90 \pm 0.15$ | $\mathbf{93.93} \pm 0.73$ | $\mathbf{93.45} \pm 1.12$ | $\mathbf{93.64} \pm 0.52$ |
| ENS(3) | $92.90 \pm 0.09$ | $88.90 \pm 0.20$ | $89.63 \pm 0.19$ | $88.16 \pm 0.20$ |
| ENS(15) | $\mathbf{93.33} \pm 0.06$ | $91.93 \pm 0.12$ | $92.83 \pm 0.11$ | $91.09 \pm 0.12$ |
| CSD | $91.93 \pm 0.17$ | $\mathbf{96.18} \pm 0.67$ | $\mathbf{96.49} \pm 0.74$ | $\mathbf{95.74} \pm 0.62$ |
| CSD-Aug. | $91.92 \pm 0.16$ | $\mathbf{97.84} \pm 0.30$ | $\mathbf{98.24} \pm 0.27$ | $\mathbf{97.34} \pm 0.31$ |
| CSD-OOD. | $91.96 \pm 0.13$ | $\mathbf{97.35} \pm 0.50$ | $\mathbf{97.87} \pm 0.45$ | $\mathbf{96.72} \pm 0.56$ |

Table 9: Distribution Shift Detection. MNIST as ID dataset.

| Method | Acc. | AUROC | AUPR-IN | AUPR-OUT |
|---|---|---|---|---|
| MCD | $98.97 \pm 0.06$ | $90.03 \pm 0.23$ | $87.70 \pm 0.38$ | $89.01 \pm 0.32$ |
| BNN-MCMC | $94.29 \pm 0.39$ | $80.24 \pm 2.19$ | $80.20 \pm 2.05$ | $77.33 \pm 2.56$ |
| BNN-Laplace | $94.17 \pm 1.01$ | $74.05 \pm 1.70$ | $72.24 \pm 1.90$ | $74.39 \pm 1.73$ |
| RND | $99.85 \pm 0.02$ | $94.66 \pm 0.52$ | $93.83 \pm 0.95$ | $\mathbf{94.25} \pm 0.35$ |
| ENS(3) | $99.95 \pm 0.01$ | $94.03 \pm 0.24$ | $95.09 \pm 0.22$ | $92.32 \pm 0.31$ |
| ENS(15) | $\mathbf{99.97} \pm 0.00$ | $\mathbf{95.33} \pm 0.06$ | $\mathbf{96.31} \pm 0.06$ | $93.79 \pm 0.10$ |
| CSD | $99.88 \pm 0.01$ | $\mathbf{96.78} \pm 0.58$ | $\mathbf{96.96} \pm 0.72$ | $\mathbf{96.25} \pm 0.57$ |
| CSD-Aug. | $99.87 \pm 0.02$ | $\mathbf{98.39} \pm 0.17$ | $\mathbf{98.63} \pm 0.20$ | $\mathbf{97.94} \pm 0.19$ |
| CSD-OOD. | $99.87 \pm 0.02$ | $\mathbf{99.37} \pm 0.08$ | $\mathbf{99.51} \pm 0.07$ | $\mathbf{99.14} \pm 0.11$ |

Table 10: Distribution Shift Detection. NotMNIST as ID dataset.

| Method | Acc. | AUROC | AUPR-IN | AUPR-OUT |
|---|---|---|---|---|
| MCD | $95.17 \pm 0.14$ | $83.21 \pm 0.45$ | $75.86 \pm 0.89$ | $85.73 \pm 0.18$ |
| BNN-MCMC | $90.20 \pm 0.44$ | $87.05 \pm 0.80$ | $85.93 \pm 1.10$ | $87.68 \pm 0.63$ |
| BNN-Laplace | $95.29 \pm 0.52$ | $86.38 \pm 1.46$ | $82.99 \pm 2.36$ | $87.55 \pm 1.04$ |
| RND | $96.25 \pm 0.12$ | $\mathbf{95.49} \pm 0.82$ | $\mathbf{95.81} \pm 0.97$ | $\mathbf{95.23} \pm 0.74$ |
| ENS(3) | $97.12 \pm 0.08$ | $92.37 \pm 0.26$ | $92.11 \pm 0.30$ | $91.93 \pm 0.27$ |
| ENS(15) | $\mathbf{97.47} \pm 0.05$ | $94.04 \pm 0.16$ | $94.26 \pm 0.17$ | $93.29 \pm 0.17$ |
| CSD | $96.48 \pm 0.08$ | $\mathbf{96.98} \pm 0.41$ | $\mathbf{97.26} \pm 0.44$ | $\mathbf{96.86} \pm 0.36$ |
| CSD-Aug. | $96.45 \pm 0.09$ | $\mathbf{98.51} \pm 0.22$ | $\mathbf{98.70} \pm 0.24$ | $\mathbf{98.31} \pm 0.21$ |
| CSD-OOD. | $96.49 \pm 0.10$ | $\mathbf{98.49} \pm 0.35$ | $\mathbf{98.78} \pm 0.29$ | $\mathbf{98.21} \pm 0.42$ |

Table 11: Distribution Shift Detection. KMNIST as ID dataset.

| Method | Acc. | AUROC | AUPR-IN | AUPR-OUT |
|---|---|---|---|---|
| MCD | $94.18 \pm 0.26$ | $87.22 \pm 0.75$ | $83.48 \pm 0.74$ | $88.00 \pm 0.77$ |
| BNN-MCMC | $80.57 \pm 1.29$ | $80.40 \pm 1.46$ | $79.31 \pm 1.93$ | $80.75 \pm 1.31$ |
| BNN-Laplace | $85.39 \pm 1.79$ | $78.58 \pm 2.66$ | $76.18 \pm 3.11$ | $79.47 \pm 2.49$ |
| RND | $96.73 \pm 0.21$ | $93.50 \pm 1.17$ | $93.58 \pm 1.45$ | $92.93 \pm 1.05$ |
| ENS(3) | $97.68 \pm 0.10$ | $93.88 \pm 0.24$ | $94.49 \pm 0.26$ | $93.05 \pm 0.24$ |
| ENS(15) | $\mathbf{97.96} \pm 0.06$ | $\mathbf{94.68} \pm 0.11$ | $\mathbf{95.39} \pm 0.12$ | $\mathbf{93.81} \pm 0.11$ |
| CSD | $96.89 \pm 0.13$ | $\mathbf{96.57} \pm 0.73$ | $\mathbf{97.05} \pm 0.74$ | $\mathbf{95.90} \pm 0.74$ |
| CSD-Aug. | $96.90 \pm 0.19$ | $\mathbf{98.12} \pm 0.46$ | $\mathbf{98.45} \pm 0.41$ | $\mathbf{97.61} \pm 0.53$ |
| CSD-OOD. | $96.86 \pm 0.12$ | $\mathbf{99.06} \pm 0.19$ | $\mathbf{99.30} \pm 0.14$ | $\mathbf{98.71} \pm 0.25$ |

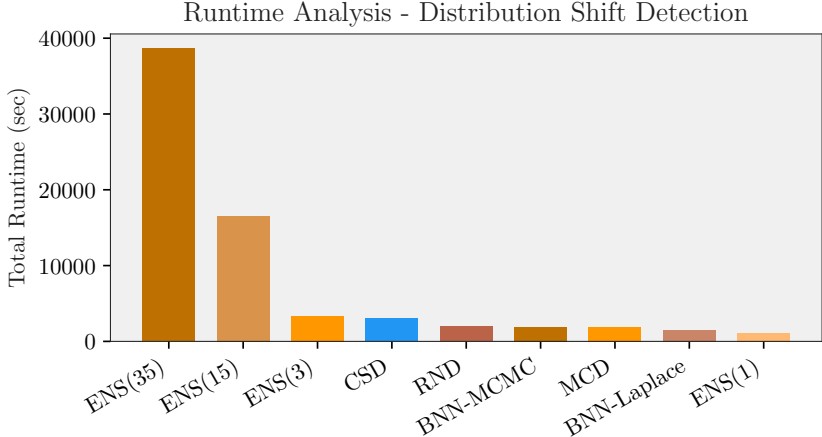

Figure 7: Runtime of all algorithms on the Distribution Shift Detection task. Runtimes are reported in seconds for one seed completion on a single Nvidia RTX 3060 GPU.

## USE OF LARGE LANGUAGE MODELS

Large language models (LLMs) were used to assist in the preparation of this paper. Their usage was limited to refining sentence structure and verifying grammar, punctuation, and general language usage. No content or substantive research contributions were generated by LLMs.

