# OpenReview forum: "Contextual Similarity Distillation: Ensemble Uncertainties with a Single Model"
_ICLR.cc/2026/Conference — ICLR 2026 Poster_

### Official Review · Reviewer_Bv9r · 2025-10-18

**Soundness:** 3
**Presentation:** 3
**Contribution:** 3
**Rating:** 8
**Confidence:** 3

**Summary:**

The authors propose  contextual similarity
distillation, an approach that explicitly estimates the variance of an ensemble
of deep neural networks with a single model, without learning or evaluating
such an ensemble in the first place. It can estimate predictive variance at inference time with a single forward pass,
and can make use of unlabeled target-domain data or data augmentations to refine
its uncertainty estimates.

**Strengths:**

The paper is clear, well-written, solves a cogent problem, and presents convincing experimental evidence.

**Weaknesses:**

The paper only has a conceptual weakness, that is widespread in the ML literature.

If we posit that epistemic uncertainty (EU) is encoded in the parameter distribution, then a single posterior predictive is not able to gauge EU correctly, since it is the expectation of the likelihood taken wrt the posterior, $p(\tilde{y} | \tilde{x},D) = \int_\Theta p(\tilde{y} | \tilde{x},\theta) p(\theta | D) \text{d}\theta = \mathbb{E}_{\theta \sim p(\theta | D)}[p(\tilde{y} | \tilde{x},\theta)]$, hence all the uncertainty encoded in the posterior gets ``washed away'' by taking the expectation.

This is not the case in ensembles, where we have different posterior predictives, but if we then distil all the models into only one, and estimate the variance of the latter thinking that it captures the EU, we incur the same problem as above. In symbols, we would have $p(\tilde{y} | \tilde{x},D) = \sum_{k=1}^K w_k \int_\Theta p_k(\tilde{y} | \tilde{x},\theta) p_k(\theta | D) \text{d}\theta = \mathbb{E}_{k} [p_k(\tilde{y} | \tilde{x},D)]$, where the $w_k$'s are the weights that the ensemble method assigns to each model $k = 1,\ldots,K$ (which of course are nonnegative and sum up to $1$).

The way to avoid this is to keep the posterior predictives separate, e.g. by considering their induced convex hull, and use other metrics existing in the literature (e.g. in https://link.springer.com/article/10.1007/s10994-021-05946-3) to quantify EU (and also aleatoric uncertainty). This was already done in https://openreview.net/forum?id=4NHF9AC5ui, and applied to active learning (thus very close to RL) in https://dl.acm.org/doi/10.1145/3716863.3718040, where EU is used to inform the exploration of the state space.

While the paper solves a very relevant problem, in a creative and innovative way, I urge the authors to (a) either avoid referring to epistemic uncertainty, and rather just call it uncertainty (in which case my previous point can be ignored), or (b) to add even a small discussion where they acknowledge my point above, and state that they keep the term "epistemic" only to conform to the existing literature.

Moreover, I suggest the authors, in future work, to look into the techniques developed by the imprecise probabilistic machine learning field, that may well turn out useful for the future endeavor pointed out in the conclusion.

**Questions:**

In line 86, shouldn't it be $\mu \in \mathscr{P}(\mathcal S)$?

---

> ### Author Response · Authors · 2025-11-19
> **Response to Reviewer Bv9r**
>
> **We thank the reviewer for their thoughtful comments and positive feedback. We appreciate the conceptual point raised regarding epistemic uncertainty, and clarify our terminology accordingly.**
>
> **On the use of the term ``epistemic uncertainty.''**
>
> We thank the reviewer for raising this nuanced point. We agree that posterior predictive distributions in Bayesian modeling typically reflect *total uncertainty*, combining both epistemic uncertainty (due to uncertainty in the posterior distribution) and aleatoric uncertainty (due to observation noise). This is particularly prevalent in GPs that model noise heteroscedastically (but also in the homoscedastic case). Such observation noise then typically shows up in the form of an additive term like $\sigma_\epsilon^2 I$ in the variance of the posterior predictive distribution. Indeed we agree with the view formulated in Huellermeyer et al. ([1], page 26), that the posterior predictive is representative of the total uncertainty, as the reviewer states. In our work, we do not model any observation noise term, i.e., our work supposes $\sigma_\epsilon^2 = 0$, and thus interpret the variance of the posterior predictive as \emph{epistemic uncertainty}. We agree with the reviewer that this distinction is subtle and have added a short clarification to the appendix B.2 to acknowledge this discussion and to provide context to our terminology.
>
> **Q1: Typo**
>
> We thank the reviewer for catching this error - we corrected it in the revised manuscript.
>
> [1] Hüllermeier, Eyke, and Willem Waegeman. "Aleatoric and epistemic uncertainty in machine learning: An introduction to concepts and methods." Machine learning 110.3 (2021): 457-506.

---

### Official Review · Reviewer_ccSm · 2025-10-27

**Soundness:** 2
**Presentation:** 2
**Contribution:** 2
**Rating:** 2
**Confidence:** 3

**Summary:**

This paper proposes contextual similarity distillation (CSD), a single-model method that tries to estimate the variance of a random initialization ensemble of DeepEnsembles by using the kernel similarities in the covariance in the Neural Tangent Model. Empirical results show CSD performs competitively and sometimes superior to ensemble-based baselines in OOD detection and sparse-reward RL environments.

**Strengths:**

- The paper is generally well-written to understand the key aspects of the proposed algorithm.
- The direction of this paper is important because DeepEnsembles' superiority in uncertainty and robustness, but struggle with computational efficiency.
- Empirical results are quantitatively shown in the experiments and qualitatively shown in Figures 1 and 2 to understand how the method actually works.

**Weaknesses:**

- Using NTK to get epistemic uncertainty is not new and has been studied in several works [1,2,3]. This limits the novelty contribution of CDS. Note that I still appreciate your contribution to the similarity distillation.
- There is no theoretical contribution in this paper to formally explain why and how CDS works.
- The experiments lack several computational efficiency baselines in uncertainty and robustness (e.g., BatchEnsembles, Rank1-BNN, etc. [4]) and OOD detection baselines (ReAct, DICE, etc. [5]).
- The main contribution is improving computational efficiency, but there is no computational efficiency evaluation (training, inference speed, GPUs, etc.) in the experiments.
- Contribution to uncertainty also needs to improve beyond OOD detection evaluation, such as Expected Calibration Error (in IID and OOD data), sharpness, etc.
- In writing, background in Reinforcement Learning, such as MDP, Q-function, policy improvement objective, are not related to the main method in Section 3 (many mathematical notations are not reused at all). Note that I still appreciate your additional RL experiments, but I would suggest moving some RL background in Section 2 to the Appendix.

**Questions:**

1. How does CDS perform on larger-scale datasets such as CIFAR-100 and ImageNet?

2. In a large-scale dataset, where the feature dimension is higher, how about the training/inference efficiency of CDS get impacted? How is the ensemble variance estimator impacted (can the authors formally show the estimator error in this regard?)?

References:

[1] He et al., Bayesian Deep Ensembles via the Neural Tangent Kernel, NeurIPS, 2020.

[2] Sergio et al., Epistemic Uncertainty and Observation Noise with the Neural Tangent Kernel, NeurIPS, 2022.

[3] Seijin et al., Disentangling the Predictive Variance of Deep Ensembles through the Neural Tangent Kernel, NeurIPS, 2022.

[4] Nado et al., Uncertainty Baselines: Benchmarks for Uncertainty & Robustness in Deep Learning, Bayesian Deep Learning workshop, NeurIPS 2021.

[5] Yang et al., OpenOOD: Benchmarking Generalized Out-of-Distribution Detection, NeurIPS 2022 Datasets and Benchmarks Track.

---

> ### Author Response · Authors · 2025-11-19
> **Response to Reviewer ccSm**
>
> **We thank the reviewer for engaging with our work and the thorough feedback. We respond to each of the raised concerns below.**
>
> **W1: Novelty.**
>
> We agree that the NTK has been explored in the context of uncertainty quantification in NNs (e.g. the works [1-3] listed by the reviewer). Our contribution is not the general use of NTK theory, but rather a novel learning algorithm (CSD) that takes inspiration from the closed-form solution to the learning dynamics of NNs in the NTK regime and frames uncertainty quantification as a similarity prediction problem. This formulation and the subsequent learning algorithm, to our knowledge, are novel.
>
> **W2: Why and how CSD works.**
>
> We provide an intuitive conceptual justification for CSD by connecting it to the behavior of deep ensembles under the NTK regime. However, our final algorithm indeed comprises several approximations and design choices that deviate from the theoretical motivation. We would like to highlight that our Appendix B: ``Discussion on Approximations'' outlines conditions, in which our proposed method *exactly* recovers the desired predictive variance of deep ensembles. In this sense, we believe our work does provide a theoretical explanation of how and why CSD works, albeit under specific conditions. Modelling the learning dynamics of the more practical implementation of CSD is a significantly nontrivial task that would warrant a worthwhile research undertaking in its own right (e.g. many papers are entirely dedicated to modelling learning dynamics of contrastive learning algorithms [1,2,3]), but lies beyond the scope of our current work.
>
> **W3: Baseline choices.**
>
> We agree that additional baselines are desirable in a general sense. However, we believe our current selection of baselines covers important families of uncertainty quantification approaches (deep ensembles, distillation, approximate Bayesian approaches) that are semantically related to our work while keeping experimental complexity tractable. In addition, we would like to underscore that the emphasis in this contribution lies not in the outperforming of all existing methods, but in offering a conceptually novel view on uncertainty estimation as a similarity prediction task and a novel learning algorithm that leverages this perspective.
>
> **W4: Computational efficiency comparison.**
>
> We thank the reviewer for this suggestion. We have added runtime comparisons on equivalent hardware to the appendix of the revised version. These show that CSD runs at significantly reduced runtime compared to full, even small ensembles while incurring slight overhead compared to a single model - a result consistent with the computational operations performed by the respective algorithms.
>
> **W5: Evaluation metrics.**
>
> We agree that several evaluation metrics are of general interest in uncertainty quantification, including calibration, sharpness, and decision-theoretic metrics. In this work, however, we focused on metrics that indicate the reliability of OOD detection (including exploration experiments) in a threshold-invariant manner, as this forms the necessary basis for downstream refinements aimed at calibration, improving sharpness or similar. We note that it is commonplace with many uncertainty quantification techniques in deep learning - including deep ensembles - to implement calibration techniques as a second step (e.g., by rescaling, remixing, or through more advanced techniques. See for example [4-6]) and even large deep ensembles are known to not be well-calibrated out-of-the-box ([4-6]). We thus consider this a valuable direction for future work.
>
> **Q1: Performance on larger-scale datasets.**
>
> We did not include CIFAR-100 or ImageNet in our evaluation. However, our RL experiments comprise image data that spans up to 10 million environment frames, which puts them closer to the large-scale setting of these datasets and show that CSD scales gracefully to this setting. Investigating the behavior of CSD in even higher-dimensional or larger vision tasks remains an exciting avenue for future work.
>
> **Q2: Effect of feature dimensionality.**
>
> Assuming the reviewer refers to the dimensionality of the feature and context vectors used in CSD, then the computational cost of the last-layer matrix multiplication scales linearly with the output dimension. This is the standard cost associated with linear layers. We are unsure which specific ``estimator error'' the reviewer is referring to; Our approach in general uses a similar algorithmic and architectural structure to many contrastive learning approaches and therefore scales similarly with input dimensionality (but highly dependent on the specific model implementation). *Formally* characterizing how approximation errors will be affected in a practical setting (i.e., deviating from the NTK regime) would require a significantly more complex analysis with strong assumptions on the data distribution and model classes and is beyond our current scope.

---

> > ### Author Response · Authors · 2025-11-19
> > **References**
> >
> > [1] Anil, Gautham Govind, Pascal Esser, and Debarghya Ghoshdastidar. "When Can We Approximate Wide Contrastive Models with Neural Tangent Kernels and Principal Component Analysis?." Proceedings of the AAAI Conference on Artificial Intelligence. Vol. 39. No. 15. 2025.
> >
> > [2] Yue, Yun, and Ziming Zhang. "Deep Contrastive Learning Approximates Ensembles of One-Class SVMs with Neural Tangent Kernels."
> >
> > [3] Tian, Yuandong, Xinlei Chen, and Surya Ganguli. "Understanding self-supervised learning dynamics without contrastive pairs." International Conference on Machine Learning. PMLR, 2021.
> >
> > [4] Rahaman, Rahul. "Uncertainty quantification and deep ensembles." Advances in neural information processing systems 34 (2021): 20063-20075.
> >
> > [5] Zhang, Jize, Bhavya Kailkhura, and T. Yong-Jin Han. "Mix-n-match: Ensemble and compositional methods for uncertainty calibration in deep learning." International conference on machine learning. PMLR, 2020.
> >
> > [6] Kuleshov, Volodymyr, Nathan Fenner, and Stefano Ermon. "Accurate uncertainties for deep learning using calibrated regression." International conference on machine learning. PMLR, 2018.

---

> > > ### Comment · Reviewer_ccSm · 2025-11-25
> > >
> > > **Q2**. Thanks for your clarification. As I understand, the ensemble variance estimator in the RHS of Eq.12 is an approximation for the LHS. Since this is an estimator, then theoretically, I would expect a formal analysis to show an estimator error, i.e., how Eq.12'RHS deviates from Eq.12's LHS? I think this will strengthen your theoretical contribution.

---

> ### Author Response · Authors · 2025-11-28
>
> We thank the reviewer for clarifying his question. We agree that quantifying the gap between the r.h.s. approximation in Eq. 12 and the true ensemble variance on the l.h.s. of Eq. 12 is desirable. We would like to note that a universal error bound for neural networks in practice (i.e. finite-width deep networks, trained with SGD, trained on arbitrary data) remains an unsolved challenge in deep learning theory. Relaxing the assumptions of our theoretical exposition, or quantifying its errors in such a general sense involves solving open problems that are at the forefront of current theoretical research [7,9,10]. As such, deriving a bound that holds generically without additional assumptions and strict conditions would significantly exceed the scope of this (or perhaps any single) paper.
>
> That being said, we were able to outline a characterization of the error terms under more specific conditions. As detailed in our derivation and Appendix B1, Eq. 12 holds exactly under the assumption of a block-diagonal kernel structure (independence between contexts) and the NTK regime (infinite width, NTK parametrization, and gradient flow). Of course, in practice, CSD operates on finite-width networks trained with discrete gradient descent. Following previous results by Lee et al. (2019) and Daniely et al. (2016), one can bound the error introduced by the simplifying assumptions of the NTK regime.
>
> For a network of width $n$ trained with gradient descent, the deviation of our estimator from the true ensemble variance scales as $\mathcal{O}(\frac{1}{\sqrt{n}})$ with high probability over random initialization.
>
> We have updated the appendix (see Appendix B.1 Discussion on Approximations) to include this outline of the characterization of error scaling and the conditions under which this result holds. We hope that the reviewer finds this addition insightful and a valuable addition to the paper.
>
> [7] Lee, Jaehoon, et al. "Wide neural networks of any depth evolve as linear models under gradient descent." Advances in neural information processing systems 32 (2019).
>
> [8] Daniely, Amit, Roy Frostig, and Yoram Singer. "Toward deeper understanding of neural networks: The power of initialization and a dual view on expressivity." Advances in neural information processing systems 29 (2016).
>
> [9] Hanin, Boris, and Mihai Nica. "Finite depth and width corrections to the neural tangent kernel." arXiv preprint arXiv:1909.05989 (2019).
>
> [10] Seleznova, Mariia, and Gitta Kutyniok. "Analyzing finite neural networks: Can we trust neural tangent kernel theory?." Mathematical and Scientific Machine Learning. PMLR, 2022.

---

### Official Review · Reviewer_ExXT · 2025-10-28

**Soundness:** 4
**Presentation:** 3
**Contribution:** 3
**Rating:** 6
**Confidence:** 3

**Summary:**

This paper proposes contextual similarity distillation (CSD), a method which approximates an ensemble of deep networks using a single model. The key idea behind this method is motivated by neural tangent kernels (NTK), which describes how similar the gradients of two inputs are with respect to the network parameters. For a network layer with infinite width, the NTK becomes deterministic and constant (allowing it to be expressed as a Gaussian process). CSD estimates the NTK by training with regression targets corresponding to kernel similarities between data points. Empirical results support theoretical claims against several baseline methods.

**Strengths:**

1. Strong theoretical grounding. The proposed method builds off of well-studied prior literature (e.g., NTK and Gaussian processes),
2. Reduced computational cost. The proposed method is able to approximate a deep ensemble using a single model, avoiding large training and inference costs. This is supported by the experimental results on tasks like OOD detection and RL navigation.
3. Novelty. The idea to use kernel regression to analytically approximate an ensemble is novel. Furthermore, the addition of a context variable to allow the method to work on arbitrary query points.

**Weaknesses:**

1. Empirical results. The experiments are limited to small scale datasets like VizDoom and FashionMNIST with well-studied, but somewhat outdated, baselines. Inclusion of larger / more complex datasets and comparison against more recent methods would considerably strengthen the paper and give more context on the scalability of CSD.

**Questions:**

1. Can the proposed idea of contextual regression be extended to capture aleatoric uncertainty as well, or is it primarily limited to estimating epistemic uncertainty?

---

> ### Author Response · Authors · 2025-11-19
> **Response to Reviewer ExXT**
>
> **We thank the reviewer for their positive assessment and for highlighting the novelty and theoretical motivation of our approach. We respond to their comments and question below.**
>
> **Empirical scope and baselines.**
>
> We chose the common image classification datasets and VizDoom as a diverse set of domains that demands reliable OOD detection while running deep neural network models at a scale that would prohibit several other theoretically well-motivated uncertainty quantification methods (e.g. full Bayesian inference, kernel GP inference, etc.). As also noted in the main text, our main emphasis lies not on outperforming state-of-the-art methods - although deep ensembles remain one of the most widely applicable and reliable models for uncertainty quantification [1] - but on demonstrating a fundamentally novel algorithmic concept by viewing uncertainty estimation through a prediction problem of similarity measures. That said, we recognize the general value of broader comparisons but view this as a direction for future work.
>
> **Q1: Aleatoric uncertainty.**
>
> The reviewer raises an interesting question. In its current form, CSD approximates the spread of an ensemble of models trained with deterministic outputs, which corresponds to epistemic uncertainty. One could incorporate aleatoric uncertainty explicitly, for example, by modeling predictive variance via heteroscedastic regression with a negative-log-likelihood (NLL) type loss. Such a regression model would exhibit different learning dynamics than the models we analyze in the NTK regime, but may be tractable. While this is outside the scope of the present work, we agree it could be a compelling extension. We added a note on this extension in the Conclusion section.
>
> [1] Gawlikowski, Jakob, et al. "A survey of uncertainty in deep neural networks." Artificial Intelligence Review 56.Suppl 1 (2023): 1513-1589.

---

### Official Review · Reviewer_8hVE · 2025-10-28

**Soundness:** 3
**Presentation:** 3
**Contribution:** 4
**Rating:** 10
**Confidence:** 3

**Summary:**

This paper presents an approach to estimate the uncertainty of a neural network prediction through contextual similarity distillation. The method is built upon the Neural Tangent Kernel theoretical results, which express a probability distribution of a neural network function after training based on the distance between a test sample and the training sample as measured by a gradient-based kernel (NTK). Using a clever choice of labels, they train a neural network to predict the uncertainty that would result of an ensemble.

**Strengths:**

This paper brings in an interesting and novel approach to uncertainty estimation. It is strongly anchored in theoretical analysis (although with several assumptions). It is also more practical than current approaches, as it requires significantly less compute, and it performs well on different datasets compared to baselines.
The paper does a good job at presenting the concept development in a pedagogical and self-contained manner, which is necessary for anyone who is not familiar with the NTK literature.
While some of the assumptions are strong, the discussion section in the Appendix about approximations is relevant and important.
I really appreciated Figure 2 - it demonstrates clearly and convincingly the method’s objective and performance.
I believe this is an impressive piece of work which could have a lasting impact on uncertainty estimation with neural networks.

**Weaknesses:**

I did not find important weaknesses in this work. Still, elements could be improved.

While the experimental results on image classification are good and convincing, I believe other types of tasks (regression, for example) could strengthen the article and open the use of this approach for a wider range of tasks.

Regression uncertainty estimation is what is actually showed in the RL environment, but very indirectly. The RL experiment is one of the least strong aspect of this paper. While it indirectly supports the claim about CSD generating good uncertainty estimations, it is not sufficiently rigorous to demonstrate the usefulness of the approach in RL. For a stronger case, more details about the implementation and other environments would be needed. More discussion too - for example, it is surprising that CSD works better than Bootstrap DQN where CSD is supposed to approximate ensembles with lower costs and Bootstrap DQN actually uses ensembles.

**Questions:**

Questions
* In section 3.1, it is said that $g$ has the same architecture as $f$, therefore $\Theta_g(x, x’) = \Theta(x, x’)$. In the text earlier, there is no justification for this: the closer argument I saw was that $\Theta(x, x’)$ is equal for every stage of training of weights $\theta_f$ under the assumption of gradient flow. Is the two networks having the same architecture sufficient for this equality to hold? If so, could you introduce this somewhere in section 2.2?

* Figure 1, the legend describes the “Kernel Prior $\Theta(x,x)$”, yet the function changes across the three subfigures, where $x_t$ is the element that changes. Should it be $\Theta(x, x_t)$?
* Section 3.3: is the feature vector $\phi(x, \tilde{\theta}_f)$ related to parameterized function $f$? If so, what exactly from $f$ does it refer to?
* Equation 14, is the RHS calculated on the trained $f$ or on the initial $f$ parameters? (This also applies to equation 13 too if $\phi(x, \tilde{\theta}_f)$ is also based on $f$).
* If it is on the trained parameters, how does that work when it is estimated during a continuous training process, such as in RL?

Comments:
* Introduction, Section 3.1: It is said often in the paper that, in the field of RL, there are often large models or datasets. Depending on the RL application, this may not be true: most applied RL uses small neural networks, and datasets are created from experience (which can be sparse/expensive). Maybe these claims could be more nuanced?
* Section 2: The objective of finding the optimal policy is not RL specific (this is the case for every MDP solving method, s.a. dynamic programming or model-predictive control). The RL specific challenge is to do so through learning on trajectories acquired in the environment.
* Section 2.1: The fundamental challenge of exploration is specific to the case of online RL (to differentiate from offline RL)
* Insufficient upper margin for eq. (3)
* End of section 2.1: Due to their initialization, ensemble members are not great at the beginning of the RL process to determine the model’s uncertainty: that’s where randomized prior functions (RPF, Osband et al., 2019) are helpful.

Typos:
* Introduction: “distribution shift detection tasks(Van Amersfoort” and “NOTMNIST datasets(Xiao et al.”: space missing before the “(“
* Section 2: “agents […] subsequently receiveS the immediate reward […] and observeS the next state”: there should not be an S as agents is plural
* Section 3.1: “it is our goal IS to estimate”: correct the sentence

---

> ### Author Response · Authors · 2025-11-19
> **Response to Reviewer 8hVE**
>
> **We thank the reviewer for their thoughtful and encouraging assessment. We appreciate their recognition of our theoretical work, its practical revelance and our presentation. Below, we respond to their questions and suggestions in turn.**
>
> **RL experiments and comparison to Bootstrapped DQN.**
> Our aim with these experiments was to assess the efficacy of CSD's epistemic uncertainty estimates in hard exploration problems. Notably, there are several reasons why CSD may outperform Bootstrapped DQN in practice: (1) the ensemble size in Bootstrapped DQN is necessarily limited by compute constraints, especially for large models, which may limit its ability to capture uncertainty reliably; (2) CSD uses uncertainty estimates as an intrinsic reward signal, whereas Bootstrapped DQN operates with an ensemble of distinct value functions. While one could modify Bootstrapped DQN to use its ensemble variances as an intrinsic reward, this would diverge from standard practice and was outside the scope of our baseline comparisons.
>
> **Q1: Equality of kernels.**
> The reviewer is correct that stating $\Theta_g(x, x') = \Theta(x, x')$ based solely on shared architecture is imprecise. What we intended to express is that the NTK $\Theta$, defined by an inner product of gradient vectors, is uniquely determined by the definition of forward/backward passes of the network and the initial weight distribution. We loosely referred to this as ``architecture''. In this context, we mean two indepedendent neural networks that use the same architecture (definition of forward / backward passes) and weight initialization distribution. We agree that this point should be explained more explicitly and we revised the wording in Section 3.1 accordingly. The reviewer is correct that this equality holds throughout training under gradient flow.
>
> **Q2: Clarification on Figure 1.**
>
> We thank the reviewer for catching this. In fact, $\Theta(x,x)$ is constant across all subplots in Figure 1. The curves are vertically shifted for visualization purposes to better highlight that the offset between the kernel prior and the similarity regression function ( $\Theta(x,x) - g_{x_t}(x,\tilde{\theta}_\infty)$ ) corresponds to the predicted variance (the length of the purple vertical bar). We revised our manuscript to include a clarification note to emphasize that the kernel prior function $\Theta(x,x)$ remains unchanged across subfigures.
>
> **Q3: Feature vector $\phi(x, \tilde{\theta}_{f})$.**
>
> The feature vector $\phi(x, \tilde{\theta_f})$ is indeed not related to the parametrized function $f$ (other than belonging to the same CSD framework that we introduced). The subscript $f$ in the feature vector is meant as a shorthand for ``feature'', not the earlier introduced function $f$. This is indeed not optimal and we renamed the parameter to $\tilde{{\theta}}_{\text{feat}}$ to clarify this distinction in the revised text.
>
> **Q4: Kernel function computation.**
> Both Equation $(13)$ and $(14)$ use the NTK computed with weights at initialization. In the infinite-width regime, the NTK remains constant during training, which justifies this choice. While using a trained network to compute this approximate kernel prior function is also possible in principle, this would introduce additional computational steps and additional complexity as the reviewer correctly pointed out.
>
> **Additional comments.**
> We appreciate the reviewer’s detailed comments on style and presentation. We corrected the wording of several statements and incorporated the reviewer's comments and identified typos in the revised manuscript. Thanks for this!

---

> > ### Comment · Reviewer_8hVE · 2025-11-25
> >
> > I thank the authors for their response! I am satisfied with the answers to my questions.
> > Needless to say, I cannot increase my rating for this paper.

---

### Meta-Review · Area_Chair_zAWU · 2025-12-30

**Summary:**

While the use of the NTK for uncertainty quantification has been extensively studied, many reviewers still find the work interesting and believe it contains valuable contributions, particularly the use of the contextual similarity distillation framework. The main concerns relate to the theoretical gaps in the methodology arising from multiple layers of approximation, the need for stronger and more comprehensive experimental settings, and the presentation of the results. The authors have made efforts to address many of the reviewers' feedbacks. Further strengthening the theoretical justification, expanding the experimental evaluation, and refining the presentation would enhance the clarity and impact of the results, but the core ideas are viewed as promising and offering substantive contributions.

**Reviewer Concerns:**

The authors have made progress in addressing reviewer concerns by providing partial theoretical justification and expanding the experimental evaluation, including additional analyses such as runtime comparisons. These efforts are appreciated. At the same time, some issues remain, both from the original reviews and those identified by the AC. In particular, further work to more tightly quantify the approximations under more general and less restrictive assumptions would strengthen the theoretical foundations. The study would also benefit from more comprehensive comparisons against a broader set of baselines (e.g., more recent NTK-based UQ methods), evaluations on larger-scale datasets and models, and additional ablation studies to better elucidate the strengths and limitations of the proposed approach across settings. In addition, the presentation of the results could be improved (e.g., the paper includes a lengthy discussion, including several notation, of MDPs in the main body that does not appear to be directly related to the core methodology).

**Reviewer Scores:**

Many reviewers have already rated the contributions of the paper highly. The reviewer who initially assigned the lowest score raised a follow-up question, to which the authors provided a partial response. While this may lead to an increase in that reviewer’s score, it is unlikely to raise it beyond a position marginally below the acceptance threshold.

---

### Decision · Program_Chairs · 2026-01-26

Accept (Poster)